# Brep2Shape: Boundary and Shape Representation Alignment via Self-Supervised Transformers

**Yuanxu Sun** [* 1]  **Yuezhou Ma** [* 1]  **Haixu Wu** [1]  **Guanyang Zeng** [1]  **Muye Chen** [1]  **Jianmin Wang** [1]  **Mingsheng Long** [1]

## Abstract

Boundary representation (B-rep) is the industry standard for computer-aided design (CAD). While deep learning shows promise in processing B-rep models, existing methods suffer from a representation gap: continuous approaches offer analytical precision but are visually abstract, whereas discrete methods provide intuitive clarity at the expense of geometric precision. To bridge this gap, we introduce Brep2Shape, a novel self-supervised pre-training method designed to align abstract boundary representations with intuitive shape representations. Our method employs a geometry-aware task where the model learns to predict dense spatial points from parametric Bézier control points, enabling the network to better understand physical manifolds derived from abstract coefficients. To enhance this alignment, we propose a Dual Transformer backbone with parallel streams that independently encode surface and curve tokens to capture their distinct geometric properties. Moreover, the topology attention is integrated to model the interdependencies between surfaces and curves, thereby maintaining topological consistency. Experimental results demonstrate that Brep2Shape offers significant scalability, achieving state-of-the-art accuracy and faster convergence across various downstream tasks. Code is available at this repository: https://github.com/thuml/Brep2Shape.

## 1. Introduction

Computer-aided design (CAD) serves as the cornerstone of modern engineering and manufacturing, providing the fundamental framework for designing everything from consumer products to spacecrafts (Heidari & Iosifidis, 2025). The industry standard for these designs, Boundary representation (B-rep), encodes geometry through a complex interplay of precise parametric geometries and topological relationships (Weiler, 1986). Traditionally, analyzing B-rep models has been a labor-intensive process requiring extensive domain expertise. Recently, deep models have emerged as promising tools for processing B-rep models. Benefiting from their impressive non-linear modeling capacity, deep models have demonstrated remarkable potential on specialized understanding tasks, such as classification and segmentation (Jayaraman et al., 2021; Zou & Zhu, 2025).

Beyond their prevalence, B-reps present unique challenges for deep models. Historically, topological relations in B-reps have been predominantly encoded through graph-based paradigms. Moreover, regarding the underlying geometry, as illustrated in Figure 1(a), existing methods generally follow two paradigms: *continuous methods* and *discrete methods*. Continuous methods, such as BRT (Zou & Zhu, 2025), leverage *boundary representations* to achieve analytical precision via parametric control points. However, these opaque inputs are often decoupled from the concrete spatial space and formalized by functions, making them hard to capture shape properties. Conversely, discrete methods, represented by UV-Net (Jayaraman et al., 2021) and BrepNet (Lambourne et al., 2021), rely on *shape representations* achieved by spatial sampling for perceptual intuition. While more accessible, they are imprecise and sensitive to discretization artifacts, sacrificing the high-fidelity essential to this domain. The representation gap remains a major obstacle to generalizable B-rep learning, raising a question: *how can we bridge this gap to learn a representation that is both mathematically rigorous and geometrically intuitive?*

To bridge this gap, we embrace a shift toward *aligned methods*, as shown in Figure 1(a). This approach is powered by *self-supervised pre-training*, mirroring its success in other domains (Brown et al., 2020; He et al., 2022). However, establishing such alignments for B-reps remains a formidable challenge. While initial efforts like SSL4CAD (Jones et al., 2023) have made preliminary strides, they often struggle with over-simplified geometries and lack the universality

---

[1]School of Software, BNRist, Tsinghua University. Yuanxu Sun <sunyuanx22@mails.tsinghua.edu.cn>. Correspondence to: Mingsheng Long <mingsheng@tsinghua.edu.cn>, Haixu Wu <wuhaixu98@gmail.com>.

*Proceedings of the 43rd International Conference on Machine Learning*, Seoul, South Korea. PMLR 306, 2026. Copyright 2026 by the author(s).

*Figure 1.* (a) Representation gap between continuous methods and discrete methods. (b) Overview of Brep2Shape. Our self-supervised pre-training task aligns *precise* expressions with *intuitive* geometries to learn *generalizable* representations for downstream tasks.

for complex B-reps. The primary barrier lies in the inherent *heterogeneity* and *abstraction* of raw B-rep data. Specifically, the varying number of control points precludes direct integration with standard deep models, and the parametric expressions remain decoupled from spatial intuition, hindering the deep model's perception of actual spatial occupancy.

Based on these insights, we propose *Brep2Shape*, a novel self-supervised pre-training method designed to align abstract boundary representations with intuitive shape representations. To ensure mathematical integrity and generalization, Brep2Shape operates directly on parametric B-rep entities. As shown in Figure 1(b), we address data heterogeneity by decomposing arbitrary entities into a sequence of standardized Bézier primitives. These primitives, defined by a fixed-size control point set, provide a structured yet continuous input space conducive to scalable pre-training. Through Brep2Shape, the model learns to map these opaque expressions into intuitive shape representations by predicting dense spatial points sampled from surfaces and curves. To support this task, we introduce *Dual Transformer*, which employs parallel streams to independently encode face and edge tokens, preserving their distinct geometric properties. We present the *topology attention* via incorporating a topological attention bias derived from the B-rep adjacency graph and its topological duality. By injecting these priors, the model maintains topological consistency and captures interdependencies between entities. Empirically, Brep2Shape demonstrates remarkable scalability and yields highly generalizable representations during pre-training, outperforming state-of-the-art task-specific models in downstream tasks with superior accuracy and faster convergence. Overall, our contributions can be summarized as follows:

- We propose Brep2Shape, a novel self-supervised pre-training method that predicts shape representations directly from boundary representations, aligning precise analytical expressions with intuitive geometries.

- Tailored to our pre-training task, we propose a Dual Transformer backbone that processes surface and curve tokens in parallel and incorporates a topology attention to capture the interdependency between B-rep entities.

- Brep2Shape demonstrates its scalability and efficacy via large-scale pre-training, achieving state-of-the-art accuracy and faster convergence on downstream tasks.

**Conflict of Interest Disclosure** Conflict of Interest Disclosure. The authors declare no financial conflicts of interest related to this work.

## 2. Related Work

### 2.1. Self-Supervised Pre-Training

The self-supervised pre-training paradigm has revolutionized deep learning by unlocking the immense potential of large-scale unlabeled datasets. In NLP and CV, models (Devlin et al., 2019; Radford et al., 2021; Ho et al., 2020) such as GPT (Radford et al., 2018; 2019; Brown et al., 2020) and DINO (Caron et al., 2021; Oquab et al., 2024; Siméoni et al., 2025) have demonstrated that pre-training on vast, non-annotated corpora allows the capture of generalizable contextual and visual representations. However, the B-rep learning community has yet to fully exploit the wealth of unlabeled CAD data residing in industrial repositories. UV-Net (Jayaraman et al., 2021) adopts a classical contrastive learning framework (Chen et al., 2020) to extract global shape-level embeddings. While effective for holistic tasks like classification, the pre-training task lacks fine-grained supervision for local geometries. Thus, the learned representations often struggle with intricate tasks such as segmentation. Jones et al. introduced a geometry-driven reconstruction task in which the model learns to rasterize each B-rep face. However, the method is restricted to B-rep models with fixed-size parameterizations and even specific surface types, failing to generalize to diverse geometries. Further, prior methods lack the flexibility to exploit the vast unlabeled datasets in industrial repositories. In contrast, *Brep2Shape* supports arbitrary B-rep models and establishes a universal alignment between opaque parametric entities and intuitive spatial shapes. By obviating the need for manual annotation, our framework enables a transition from task-specific learning to a scalable, self-supervised paradigm that captures robust and generalizable geometric representations.

## 2.2. Learning on B-reps

B-rep is the de facto standard for 3D modeling in engineering (Ansaldi et al., 1985). However, unlike those discrete data domains (Guo et al., 2025; Kirillov et al., 2023; Wu et al., 2024b; Zhou et al., 2025; Ma et al., 2025), B-reps represent 3D models through continuous parametric entities and topological relations, posing significant challenges for standard deep learning paradigms (He et al., 2016; Brown et al., 2020). Therefore, prior works focus on developing specialized backbones for this domain, aiming to seamlessly integrate geometric precision with topological connectivity.

Recent works on direct B-rep learning commonly rely on GNNs to model topology, following the paradigm introduced by Cao et al.. While their initial work was restricted to planar faces, subsequent methods have sought to accommodate more general surface types. This is typically achieved by those discrete methods, discretizing parametric geometry into 2D grids (Jayaraman et al., 2021; Colligan et al., 2022). Beyond surface discretization, several works incorporate handcrafted attributes, such as face types, face areas, and edge length to enrich node representations, including BRep-GAT (Lee et al., 2023), BRepMFR (Zhang et al., 2024), and AAGNet (Wu et al., 2024a). While effective, these approaches often depend on pre-discretization or domain-specific representations, which require careful tuning and may limit their generality across different B-reps.

Further, Transformers (Vaswani et al., 2017), as a vital cornerstone of deep learning, have been applied to learn on B-reps. BRT (Zou & Zhu, 2025) near-losslessly encodes entities into Bézier primitives and leverages a Transformer to model their relations, while concurrently employing a hierarchy of RNNs and Transformers to capture the multi-level topological constraints. However, the complex hierarchy may limit its scalability (Kaplan et al., 2020). In contrast, our framework adopts a pure-transformer backbone, augmenting the topology attention to capture interdependencies between faces and edges, offering superior expressivity.

## 2.3. Geometric Representations for B-reps

Geometric modeling primarily employs Bézier, B-splines and non-uniform rational B-splines (NURBS) to define curves and surfaces, offering varying levels of flexibility and precision (Patrikalakis & Maekawa, 2002). Bézier entities are defined by control points and are suitable for local geometric primitives. Moreover, B-splines generalize Bézier entities by combining several Bézier entities via knot vectors, offering better flexibility and locality. However, as polynomials, B-splines cannot exactly represent complex shapes like conic sections. Further, NURBS extend B-splines by introducing rational weights, enabling them to represent nearly all types of geometries, making them an industrial standard for high-fidelity modeling. However, the

complexity of NURBS, with varying control points, knot vectors, and weights, makes them difficult to use directly in deep models. Therefore, we decompose NURBS into a sequence of Bézier primitives with a fixed degree, enabling consistent processing across different B-rep entities (Piegl & Tiller, 2012). More details can be found in Appendix C.

## 3. Method

To bridge the gap between abstract analytical expressions with intuitive geometries, we introduce Brep2Shape, a self-supervised task that maps boundary representations to shape representations. Leveraging a novel Dual Transformer backbone that injects topological priors, Brep2Shape yields representations that align analytical fidelity with the semantic flexibility required for various downstream tasks.

**Problem setup** Formally, a B-rep model is defined as a tuple $\mathcal{B} = (\mathcal{G}, \mathcal{E})$, where $\mathcal{G}$ denotes the topological structure and $\mathcal{E}$ represents the geometric entities. The geometric entity set $\mathcal{E} = \{f_1, \ldots, f_{N_f}, e_1, \ldots, e_{N_e}\}$ is composed of $N_f$ surfaces and $N_e$ curves, each parameterized as a NURBS entity. The topology $\mathcal{G}$ is represented by a *face graph* that captures the adjacency and boundary relationships between faces and edges (e.g., which edges bound a specific face). Our objective is to learn generalizable representations that integrate geometric precision and topological context.

### 3.1. Brep2Shape

**Boundary representaion** The inherent heterogeneity of NURBS data, with varying control points, knot vectors, and weights, makes them difficult to use directly in deep models. Therefore, we follow established geometric processing protocols and decompose each geometry entity into a sequence of Bézier primitives, using $n_f$ Bézier triangles for surfaces and $n_e$ Bézier segments for curves, each with a fixed number of control points $n$. Each control point is represented by its spatial coordinates and associated NURBS weight, producing a structured control point set $\{\mathcal{P}^i_{f_j}\}^{n_f}_{i=1}$ for the $j$-th face and $\{\mathcal{P}^i_{e_k}\}^{n_e}_{i=1}$ for the $k$-th edge, where $\mathcal{P}^i_x \in \mathbb{R}^{n \times 4}$ denotes the control point matrix of the $i$-th primitive within entity $x$. Formally, we define $\mathcal{P}$ as the representation encompassing all primitives derived from all entities, $\mathcal{P} = \text{Decompose}(\mathcal{E}) = \{\{\mathcal{P}^i_{f_j}\}^{n_f}_{i=1}\}^{N_f}_{j=1} \cup \{\{\mathcal{P}^i_{e_k}\}^{n_e}_{i=1}\}^{N_e}_{k=1}$.

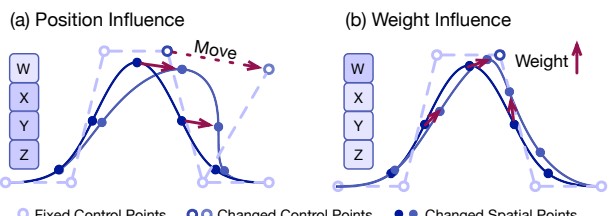

*Figure 2.* (a) Position and (b) weight modifications of a single control point lead to opaque spatially-varying shape deformations.

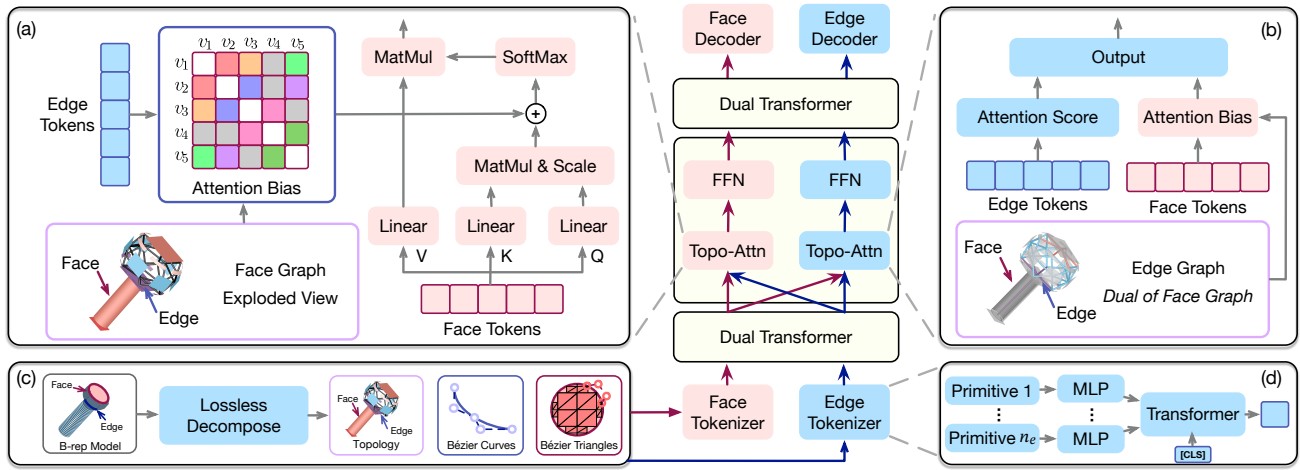

*Figure 3.* Overview of Dual Transformer. (a-b) Topology Attention: Face and edge streams incorporate cross-stream geometric features as attention biases to encode topological adjacency. (c) Decomposition: Entities in B-rep models are near-losslessly decomposed into Bézier primitives. (d) Tokenization: Hierarchical aggregation of Bézier primitives into entity-level embeddings (face or edge tokens).

While this decomposition preserves mathematical exactness, as shown in Figure 2, the resulting control point set is not an explicit geometric description. Unlike point clouds or meshes, it should be interpreted as the coefficients of a polynomial basis rather than points lying on the geometry. The final entity is obtained after complex local basis combination and global weight-induced normalization. Consequently, these control points remain fundamentally opaque, making them largely decoupled from direct geometric intuition.

**Shape representation**    We introduce a spatial point-based *shape representation* derived from the discretization of the parametric domain. Viewed as a continuous mapping, a B-rep *entity* projects a structured parametric domain into 3D space. Therefore, we perform uniform sampling within the parameter domain, evaluating the analytical mapping to yield 3D coordinates $\mathcal{U}_{f_j}$ for the $j$-th face and $\mathcal{U}_{e_k}$ for the $k$-th edge. Here, $\mathcal{U}_x \in \mathbb{R}^{m \times 3}$ denotes a fixed-size spatial point set of the entity $x$, where $m$ is achieved via padding for both faces and edges. This representation serves as an explicit geometric counterpart to the opaque control points, facilitating a more accessible understanding of the entity's global shape and spatial occupancy. Formally, we define $\mathcal{U}$ as the representation encompassing all primitives derived from all entities, $\mathcal{U} = \text{Sample}(\mathcal{E}) = \{\mathcal{U}_{f_j}\}_{j=1}^{N_f} \cup \{\mathcal{U}_{e_k}\}_{k=1}^{N_e}$.

**Pre-training task**    Based on the two complementary geometric representations, $\mathcal{P}$ and $\mathcal{U}$, we propose Brep2Shape as our core pre-training task. This task aims to learn the mapping from an abstract boundary representation to an intuitive shape representation, which can be formalized as

$$\min_\theta \|\mathcal{U} - f_\theta(\mathcal{P}, \mathcal{G})\|^2, \qquad (1)$$

where $f_\theta$ denotes the deep model and $\mathcal{G}$ represents the B-rep topology. Since the shape representation can be directly obtained through analytical geometric evaluation, this task

requires no additional manual annotations and is therefore well-suited for self-supervised pre-training. Through this alignment, the model is forced to capture the complex relations between B-rep parameters and the geometries, thereby learning aligned representations with enhanced geometric intuition and generalization across downstream tasks.

*Remark* 3.1 (**Why topology matters for Brep2Shape**). Topology encodes the boundary constraints and connectivity priors between entities beyond the local control points. In Brep2Shape, incorporating topology ensures that the model does not merely learn a collection of disjoint surfaces, but understands the *geometric continuity* required at shared interfaces. Message passing across adjacent faces and edges further reconciles local analytic detail with global intuition.

*Remark* 3.2 (**Edge-level supervision as a cross-entity consistency signal**). In B-rep models, an edge serves as more than a 1D entity; it acts as a natural anchor for geometric consistency between adjacent faces. While face-level supervision aids in learning local manifold geometry, it is edge supervision that ensures spatial alignment at shared boundaries. In contrast, many existing methods focus solely on face-level supervision (Jones et al., 2023), and lack explicit information flow from faces to edges (Zou & Zhu, 2025), making it challenging to capture consistency reliably.

### 3.2. Dual Transformer for B-reps

Realizing effective alignment between boundary representation and shape representation necessitates a backbone that can capture the intrinsic relations between B-rep entities. Building upon encoding B-rep model into continuous tokens, we propose *Dual Transformer* designed to maintain topological consistency across different representations.

**B-rep tokenization**    The structural heterogeneity of B-rep entities precludes direct, uniform tokenization. To ensure

a consistent input dimension, we follow BRT (Zou & Zhu, 2025) by decomposing each entity into a sequence of Bézier primitives, utilizing $n_f$ primitives for surfaces and $n_e$ primitives for curves. Each Bézier primitive is initially projected into a latent space via an MLP. Subsequently, two independent transformer encoders are employed to model the intra-entity dependencies for faces and edges, respectively. By incorporating a learnable [CLS] token, the model aggregates the entire primitive sequence into a global entity embedding. This process can be formalized as,

$$\mathbf{x}_{f_j}^0 = \text{Transformer}_{\theta_f}([[\text{CLS}], \{\text{MLP}_{\phi_f}(\mathcal{P}_{f_j}^i)\}_{i=1}^{n_f}])_{[\text{CLS}]},$$
$$\mathbf{x}_{e_k}^0 = \text{Transformer}_{\theta_e}([[\text{CLS}], \{\text{MLP}_{\phi_e}(\mathcal{P}_{e_k}^i)\}_{i=1}^{n_e}])_{[\text{CLS}]},$$

where $\mathcal{P}^i$ denotes the control point set of the $i$-th primitive, $j \in \{1, ..., N_f\}$ and $k \in \{1, ..., N_e\}$. Finally, the $\mathbf{x}_{f_j}^0$ and $\mathbf{x}_{e_k}^0$ serve as the inputs to the Dual Transformer backbone.

**Dual transformer backbone**   Beyond individual geometries, a B-rep model is defined by the topological relations that bind separate entities into a coherent solid. Building upon those entity tokens, we propose *Dual Transformer* to explicitly encode topological dependencies. To maintain scalability, Dual Transformer largely adheres to the standard transformer while bifurcating into two parallel streams, dedicated to processing face and edge tokens, respectively.

Within each stream, the standard self-attention is replaced by our proposed *topology attention* module that injects topological priors as dynamic attention biases, as illustrated in Figure 3. In the face stream, we construct a *face graph* $\mathcal{G}^f$ where nodes (faces) are connected via shared B-rep edges. The value of attention bias between two face tokens is explicitly derived by linearly projecting the embedding of their shared edge token. This mechanism guides the model to prioritize interactions between topologically adjacent faces. Symmetrically, the edge stream utilizes an *edge graph* $\mathcal{G}^e$, i.e., the dual of the face graph, where B-rep edges act as nodes linked by shared faces. This inter-stream coupling establishes a bidirectional flow of information, allowing the model to exploit the intrinsic duality of B-rep topology. Therefore, the backbone preserves fine-grained local geometry while remaining globally context-aware. For clarity, we summarize the process as Topo-Attn$^f(\mathbf{x}_f, \mathbf{x}_e)$ for face tokens and Topo-Attn$^e(\mathbf{x}_e, \mathbf{x}_f)$ for edge tokens, respectively.

**Overall design**   In summary, *Dual Transformer* facilitates a unified representation of B-rep models by integrating explicit topological priors into a scalable transformer backbone. Suppose there are $L$ layers, as shown in Figure 3, the $l$-th layer of Dual Transformer can be formalized as follows:

$$\hat{\mathbf{x}}_\star^l = \text{Topo-Attn}^\star\left(\text{LN}(\mathbf{x}_\star^{l-1}), \mathbf{x}_{\bar\star}^{l-1}\right) + \mathbf{x}_\star^{l-1},$$
$$\mathbf{x}_\star^l = \text{FeedForward}(\text{LN}(\hat{\mathbf{x}}_\star^l)) + \hat{\mathbf{x}}_\star^l, \quad (2)$$

where $l \in \{1, \cdots, L\}$ and $\mathbf{x}_f^l \in \mathbb{R}^{N_f \times C}$, $\mathbf{x}_e^l \in \mathbb{R}^{N_e \times C}$ are outputs of the $l$-th layer. Further, $\star \in \{f, e\}$ denotes the current stream (face or edge) and $\bar\star$ denotes the complementary stream providing the attention bias. For *pre-training*, we append task-specific MLP heads to the final-layer representations $\mathbf{x}_f^L$ and $\mathbf{x}_e^L$ to reconstruct the shape representation (i.e., $\mathcal{U}_{f_j}$ for the $j$-th face). For *downstream fine-tuning*, the pre-training heads are replaced by task-specific layers to produce final predictions for diverse CAD applications.

# 4. Experiments

We conduct experiments to evaluate the scalability and transferability of Brep2Shape. The pre-trained models are fine-tuned and evaluated on representative downstream tasks.

**Pre-training data**   We collected and curated *Brep2Shape-250k*, consisting of 250,000 B-rep models from multiple public and industrial sources. The dataset integrates several prominent benchmarks (Wu et al., 2021; Lambourne et al., 2021; Cao et al., 2020). To further enhance geometric diversity, we incorporate FabWave (Angrish et al., 2019) and TMCAD (Zou & Zhu, 2025) for their complex manufacturing geometries. Notably, Brep2Shape is trained solely on raw B-rep geometry, independent of any downstream task labels, making Brep2Shape-250k serve as a scalable foundation. More details can be found in Appendix A.1.

**Benchmarks**   As presented in Table 1, four widely-adopted benchmarks are selected for downstream evaluation, covering two main tasks. For the segmentation task, we utilize MFCAD++ (2022) for machining feature segmentation, where faces are labeled according to machining features such as passages and slots. Fusion360Seg (2021) is used for semantic segmentation, labeling each face according to its modeling operation sequence, including extrude side, extrude end, and so on. For the classification task, FabWave (2019) focuses on fine-grained categorization across engineering parts, covering brackets and o-rings. TM-CAD (2025) focuses on industrial mechanical components, particularly standard parts such as bearings and flanges.

*Table 1.* Summary of datasets used for downstream tasks. Seg. and Cls. are short for segmentation and classification. * indicates that only valid files are retained from the original dataset.

| Attributes | MFCAD++ | Fusion360 | FabWave | TMCAD* |
|---|---|---|---|---|
| Tasks | Seg. | Seg. | Cls. | Cls. |
| Categories | 25 | 8 | 45 | 10 |
| Mean Faces | 30 | 15 | 22 | 75 |
| Mean Edges | 157 | 70 | 111 | 400 |
| Data Size | 59,665 | 35,858 | 4,572 | 7,599 |

**Implementations**   During pre-training, Brep2Shape is optimized using AdamW (2019) with MSE loss for 100 epochs on an NVIDIA A100 GPU. In the fine-tuning stage, we switch to cross-entropy loss and train for another 100 epochs

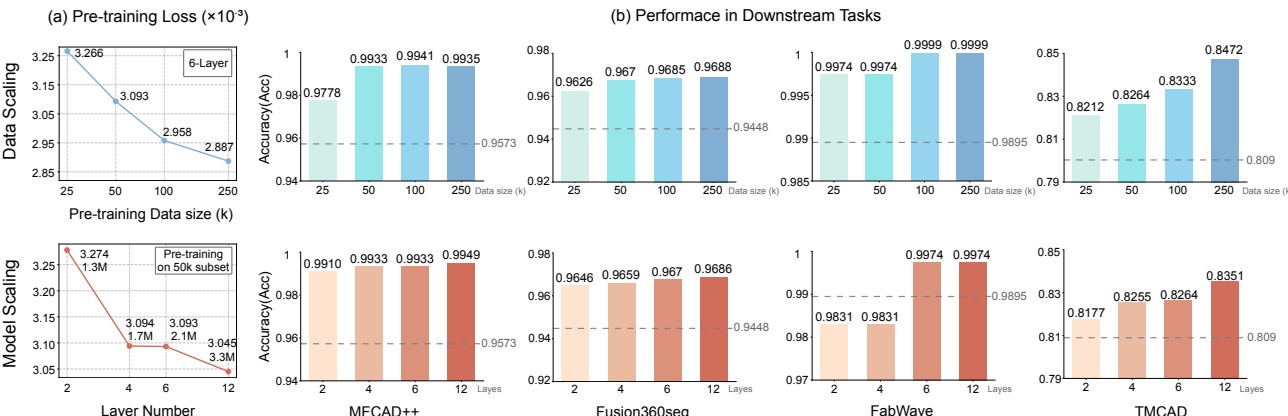

*Figure 4.* Scaling behavior of Brep2Shape. Top: Scaling pre-training data from 25k to 250k with a fixed 6-layer Dual Transformer. Bottom: Scaling model size from 2 to 12 layers using a fixed 50k uniform subset. Dashed lines denote the performance of the state-of-the-art BRT baseline. Overall, increasing data scale or model size leads to consistent improvements in pre-training and downstream tasks.

with the same optimizer. All baselines are trained for 350 epochs to adhere to their original settings. Performance is evaluated using Accuracy (Acc) for classification tasks, and both Accuracy and Intersection over Union (IoU) for segmentation tasks. See Appendix A for more details.

### 4.1. Scaling Performance

Scalability is an important property for representation learning. To investigate the scaling behavior of Brep2Shape, we pre-train it with increased data scale and model size, and evaluate their fine-tuned performance on downstream tasks.

**Data scaling** We investigate data scalability by pre-training Brep2Shape on increasingly larger subsets randomly sampled from the Brep2Shape-250K. In Figure 4(a), the pre-training loss decreases as the data scale grows, with a huge reduction of 11.6%. We then fine-tune the models on four downstream tasks. To avoid data leakage, no samples from the downstream task test sets are included in the pre-training corpus. As shown in Figure 4(b), downstream performance consistently improves with increasing pre-training data, with the average accuracy rising from 0.9379 to 0.9524, indicating effective transfer from pre-training. This data scaling behavior arises from our self-supervised objective, which enforces alignment between boundary representations and shape representations, jointly constrained by geometric coordinates and topological connectivity. Due to the diverse entities and rich compositional structure of B-rep models, increasing the pre-training data introduces novel information rather than redundant samples. This continually expands the effective supervision space, prevents early saturation, and enables consistent performance gains with larger data scale.

**Model scaling** Beyond data scaling, we progressively increase the number of Dual Transformer layers from 2 to 12, expanding the model parameters from 1.3M to 3.3M

while fixing the pre-training dataset to 50k samples uniformly drawn from Brep2Shape-250K. As shown in Figure 4, increasing model capacity consistently reduces the pre-training loss and yields improvements on downstream tasks. This suggests that the Dual Transformer can effectively utilize additional capacity, benefiting from its scalable architecture with the topology attention that promotes structured information exchange between surfaces and curves.

To summarize, scaling both the pre-training data and model layers consistently reduces pre-training loss and leads to steady downstream gains, indicating that Brep2Shape exhibits favorable scaling behavior and can effectively leverage increased data and parameters to learn higher-quality geometric representations across diverse B-rep models, surpassing the state-of-the-art baseline, BRT, by a clear margin, with an average error reduction of up to 36.3%.

### 4.2. Main Results

We compare Brep2Shape with several advanced baselines, including UV-Net, AAGNet, and BRT, across two widely-adopted tasks. For Brep2Shape, we utilize a 6-layer Dual Transformer pre-trained on the whole Brep2Shape-250k. To evaluate the efficiency and transferability of the pre-trained representations, we fine-tune them for only 100 epochs, compared to 350 epochs used for training the baselines.

**Classification** Table 2 shows that Brep2Shape outperforms prior methods on the two classification benchmarks, FabWave and TMCAD. Moreover, discrete methods such as UV-Net and AAGNet perform reasonably well on simpler datasets like FabWave, but generalize poorly to more complex geometries such as TMCAD. By aligning abstract expressions with intuitive geometries as auxiliary guidance, Brep2Shape captures fine-grained geometric details together with the topological relationship among B-rep entities. Our

*Table 2.* Fine-tuning performance comparison of Brep2Shape. The best results are highlighted in **bold** and the second best is underlined. For classification, we report Acc; for segmentation, we report segmentation Acc and IoU.

| Method | FabWave | TMCAD | MFCAD++ | | Fusion360Seg | |
|---|---|---|---|---|---|---|
| | | | Acc | IoU | Acc | IoU |
| UV-Net (2021) | 92.68 | 77.87 | 98.92 | 96.56 | 89.03 | 66.47 |
| AAGNet (2024a) | 96.33 | 74.72 | 99.29 | **98.64** | 82.45 | 75.53 |
| BRT (2025) | 98.95 | 80.90 | 95.73 | 89.98 | 94.48 | 79.23 |
| **Brep2Shape** | **99.99** | **84.72** | **99.35** | 98.02 | **96.88** | **83.77** |

learned representations demonstrate strong transferability, particularly on the more complex dataset TMCAD that features higher geometric complexity and varied topologies. In these challenging scenarios, Brep2Shape consistently outperforms existing methods, confirming its efficacy in handling sophisticated industrial components.

**Segmentation** On semantic segmentation benchmarks, Brep2Shape achieves superior performance on both MF-CAD++ and Fusion360Seg under Acc and IoU metrics, as shown in Table 2. Besides, AAGNet performs competitively on MFCAD++, benefiting from its handcrafted geometric features, but degrades on the more complex Fusion360Seg, while BRT shows the opposite trend. Benefiting from pre-training, Brep2Shape combines the precision of boundary representations with the intuition of shape representations, which is well suited for these face-level segmentation tasks. Further, since face semantics depend on geometric shape and the topological relations among neighboring faces (e.g., faces originating from the same extrusion operation), incorporating topology during pre-training enables Brep2Shape to generalize more effectively across these benchmarks.

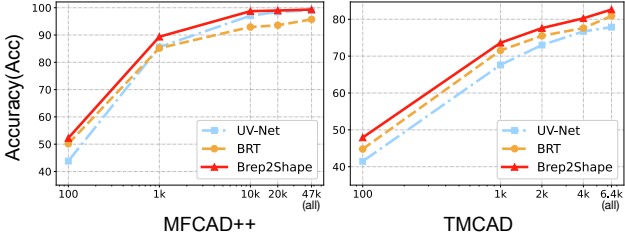

*Figure 5.* Limited data generalization analysis on MFCAD++ and TMCAD under varying numbers of labeled training samples.

**Limited data generalization** We analyze the generalizability of Brep2Shape by fine-tuning the pre-trained model with *limited* labeled downstream data in Figure 5. Brep2Shape adopts a 6-layer Dual Transformer pre-trained on a 50k subset and is fine-tuned for 100 epochs. Experiments are conducted on MFCAD++ and TMCAD. With only 100 labeled samples, Brep2Shape achieves 47.9% accuracy, compared to 41.5% for UV-Net and 44.8% for BRT, indicating that Brep2Shape representations transfer effectively to

downstream tasks. As labeled data increases, all methods benefit from additional supervision; however, Brep2Shape maintains stable performance and clear gains, suggesting it learns representations that are robust in low-label regimes. See Appendix B.2 for the numerical results of Figure 5.

## 4.3. Model Analysis

**Pre-training strategy ablations** We conduct detailed ablations to analyze different pre-training strategies. As shown in Table 3, pre-training the two tokenizers only while removing the Dual Transformer leads to a much higher pre-training loss, indicating that interactions between different entities are critical for effective B-rep learning. We further evaluate the role of edge-level supervision by removing supervision signals on edges, which leads to a clear increase in face-level loss. By enforcing spatial consistency across adjacent faces, edge-level supervision facilitates more coherent geometric and topological learning during pre-training. Full downstream results for these pre-training strategy ablations are provided in Appendix B.3, where the default Brep2Shape consistently achieves the best transfer performance.

*Table 3.* Ablations on different pre-training strategies. *The pre-training loss is decomposed into face-level and edge-level components*, both reported in units of $10^{-3}$. Dual Trm. w/o Edge-Level Supervision denotes removing supervision signals on edges.

| Method | Face Loss | Edge Loss |
|---|---|---|
| Pre-train Tokenizers Only | 2.142 | 1.287 |
| Dual Trm. w/o Edge Level Supervision | 1.872 | – |
| **Brep2Shape (Default)** | **1.855** | **1.238** |

**Finetuning strategy ablations** We evaluate different fine-tuning strategies for the pre-trained model, including linear probing, partial fine-tuning, and full fine-tuning. As shown in Table 4, linear probing achieves competitive performance on TMCAD, indicating that the pre-trained representations are already highly transferable to downstream classification tasks, while its performance on MFCAD++ is limited due to the larger number of categories. Further fine-tuning the topological encoder (Partial FT) yields performance comparable to full fine-tuning, suggesting that complex interactions are better captured with the assistance of topological modeling.

*Table 4.* Ablation study on different fine-tuning strategies. *Linear Probing* freezes the pre-trained backbone and trains only a task-specific prediction head. *Partial Fine-tuning (Partial FT)* fine-tunes the Dual Transformer while keeping the tokenizers frozen. *Full Fine-tuning (Full FT)* updates all model parameters.

| Method | TMCAD | MFCAD++ (Acc) | MFCAD++ (IoU) |
|---|---|---|---|
| Linear Probing | 0.7915 | 0.8465 | 0.6601 |
| Partial FT | 0.8168 | 0.9930 | 0.9783 |
| **Full FT** | **0.8264** | **0.9934** | **0.9799** |

**Backbone ablations** To validate the effectiveness of the Dual Transformer, we conduct ablations by varying backbone designs, including alternative cross-modal fusion strategies. Details of variants and results are shown in Table 5. As expected, GNN yields the weakest performance, suggesting that local message passing alone is insufficient for capturing global B-rep geometry. Face Trm. removes the edge stream and therefore loses boundary cues, while Dual Trm. w/ Std. Attn. keeps the dual-stream architecture but removes topology attention, leading to weaker segmentation performance. We further compare two stronger fusion variants. Face + Edge Trm. merges face and edge tokens into a single Transformer, and Q-Former introduces explicit cross-attention between the two streams. Although these variants achieve competitive classification accuracy on TMCAD, they underperform on MFCAD++ segmentation. This contrast suggests that classification mainly benefits from holistic shape-level aggregation, whereas segmentation requires preserving entity-specific face and edge semantics for precise local prediction. Thus, overly entangling face and edge features can blur geometric properties that are critical for face-level prediction. In contrast, the default Dual Transformer preserves separate face and edge streams while exchanging information through topology-aware attention biases, achieving the best overall performance.

*Table 5.* Ablations on backbones. Trm. denotes Transformer. Q-Former fuses face and edge streams via query-based cross-attention. Face Trm. only keeps the face stream. Face + Edge Trm. treats faces and edges as nodes in a single Transformer over a complete graph. Dual Trm. w/ Std. Attn. uses standard attention rather than topology attention. Pre-train loss is reported in units of $10^{-3}$. Accuracy / IoU are reported for MFCAD++.

| Backbone | TMCAD | MFCAD++ | Pre-train Loss |
|---|---|---|---|
| GNN | 0.8099 | 0.9836 / 0.9522 | 3.855 |
| Q-Former | 0.8244 | 0.9863 / 0.9603 | 3.158 |
| Face Trm. | 0.8182 | 0.9907 / 0.9689 | 3.202 |
| Face + Edge Trm. | 0.8248 | 0.9818 / 0.9487 | 3.251 |
| Dual Trm. w/ Std. Attn. | 0.8177 | 0.9853 / 0.9582 | 3.203 |
| **Dual Trm. (Default)** | **0.8264** | **0.9934 / 0.9799** | **3.093** |

**Fine-tuning Dynamics** Table 6 illustrates the fine-tuning dynamics of Brep2Shape on TMCAD under different training epochs. The accuracy increases consistently from 82.29% at 50 epochs to 84.03% at 350 epochs, indicating that the pretrained representation can be progressively adapted to the downstream classification task through longer optimization. Notably, the performance does not saturate quickly after the default 100-epoch setting, but continues to improve with additional fine-tuning. This suggests that fine-tuning mainly refines task-specific decision boundaries while preserving the general geometric knowledge learned during pretraining. The default configuration of 100 epochs achieves 82.64% accuracy, offering a practical balance between efficiency and effectiveness, whereas extending training to 350 epochs brings a further 1.39 percentage-point

improvement. The monotonic improvement also indicates stable optimization behavior, with no clear sign of overfitting in the tested range. Overall, the results show that Brep2Shape is robust to the fine-tuning schedule and can further benefit from longer downstream adaptation.

*Table 6.* Effect of fine-tuning epochs on TMCAD accuracy (Acc). The default fine-tuning for Brep2Shape adopted in all other experiments (100 epochs) is highlighted in **bold**.

| #Epoch | 50 | **100** | 150 | 200 | 250 | 300 | 350 |
|---|---|---|---|---|---|---|---|
| TMCAD | 82.29 | **82.64** | 82.99 | 83.16 | 83.33 | 83.68 | 84.03 |

**Domain transfer** Domain transfer evaluates whether models trained on one dataset can generalize to another dataset. To this end, we compare Brep2Shape with SSL4CAD (Jones et al., 2023), a pre-training method for domain transfer. Both methods are pre-trained on Fusion360Seg and fine-tuned on Fusion360Seg and MFCAD; Brep2Shape is additionally pre-trained on a 25k subset to match the scale of Fusion360Seg. As shown in Table 7, Brep2Shape achieves strong in-domain performance on Fusion360Seg and outperforms SSL4CAD on the target domain MFCAD. The improvement is most pronounced in the low-data regime, with an absolute gain of about 15% using 100 samples, and remains stable with more labeled data. With the 25k subset pre-training, Brep2Shape reaches comparable in-domain performance to SSL4CAD while maintaining a clear advantage on MFCAD, suggesting that the learned representation captures shared geometric and topological patterns rather than dataset-specific cues.

*Table 7.* Domain transfer results on Fusion360Seg (in domain) and MFCAD (cross domain). Brep2Shape-Fusion is pre-trained on Fusion360Seg and Brep2Shape-25k uses a 25k pre-training subset.

| Task / Model | Training Set Size / Accuracy | | | |
|---|---|---|---|---|
| Fusion360Seg | 1k | 10k | 20k | 23k (all) |
| SSL4CAD (2023) | 0.91 | 0.95 | 0.96 | 0.96 |
| Brep2Shape-25k | 0.8928 | 0.9518 | 0.9625 | 0.9626 |
| Brep2Shape-Fusion | **0.9201** | **0.9604** | **0.9711** | **0.9716** |
| MFCAD | 100 | 1k | 10k | 14k (all) |
| SSL4CAD (2023) | 0.66 | 0.96 | 0.99 | 0.99 |
| Brep2Shape-25k | **0.7626** | **0.9957** | **0.9999** | **0.9999** |
| Brep2Shape-Fusion | 0.7362 | 0.9946 | **0.9999** | **0.9999** |

**Representation analysis** In Figure 6(a), Brep2Shape accurately predicts planar coordinates, while curved surfaces are more challenging due to higher abstraction. Despite minor deviations, the overall shape is well reconstructed. Figure 6(b) presents t-SNE visualizations of different representations using the top-3 most frequent classes in MFCAD++. Compared to boundary representations, Brep2Shape shows clearer separation aligned with geometric intuition, leading to improved discriminability for downstream tasks. To complement the qualitative t-SNE visualization, we conduct two

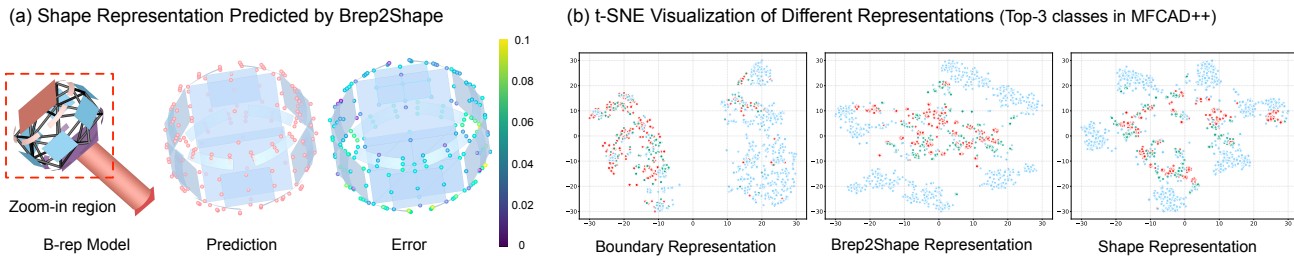

*Figure 6.* (a) Exploded view of the screw (upper portion) is shown, including the raw B-rep model, the predicted shape representation, and the corresponding point-wise error map. (b) Each point represents a face and is colored according to the top-3 classes in MFCAD++.

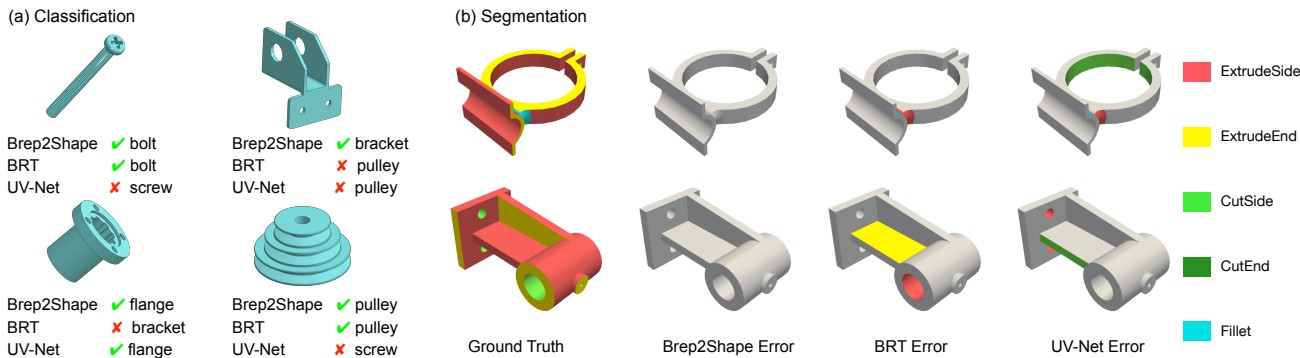

*Figure 7.* Case study on different models. (a) Classification results on *TMCAD*. (b) Segmentation results on *Fusion360Seg*. Gray regions denote correct face predictions, while colored regions mark faces predicted incorrectly. See Appendix E for more case studies.

quantitative analyses in Table 8. First, for cosine similarity analysis, we randomly select 20 faces and compute their cosine similarity to all other faces. For each selected face, we compare the label agreement rate among the top-10% most similar and bottom-10% least similar faces, and report the difference as the discriminability score. Second, for linear probing, we freeze each representation and train a single-layer MLP classifier. Brep2Shape achieves the highest discriminability score and linear probing accuracy, confirming that the proposed alignment objective yields more semantically meaningful feature spaces.

*Table 8.* Quantitative representation analysis on MFCAD++. Cosine similarity reports the label-agreement gap between the top-10% most/least similar faces, while Linear Probe Acc. measures frozen-representation face-label accuracy with a single-layer MLP.

| Method | Cosine similarity | Linear probing |
|---|---|---|
| Boundary Representation | 0.14 | 0.7035 |
| Shape Representation | 0.21 | 0.7353 |
| **Brep2Shape Representation** | 0.23 | **0.8465** |

**Fine-tuning showcases**  Figure 7 showcases the downstream performance of different models. For classification, Brep2Shape correctly recognizes these complex geometric structures, including bolts, brackets, flanges, and pulleys. These cases suggest that Brep2Shape can integrate structural cues beyond isolated local patterns. For example, UV-Net captures the local threaded geometry in the bolt case, but still misclassifies it as a screw, indicating that the distinction re-

quires reasoning over more distant regions such as the head and bottom structure. Similarly, recognizing brackets and flanges depends on understanding the global arrangement of holes, supports, and cylindrical or planar components. This shows that Brep2Shape benefits from preserving both local geometric details and long-range topological dependencies. For segmentation, Brep2Shape also shows stronger capability in fine-grained geometric understanding. As shown in the top-row examples, BRT and UV-Net make errors around small adjacent faces and subtle boundary regions, where accurate prediction requires not only local surface recognition but also contextual reasoning over neighboring entities. The fewer errors produced by Brep2Shape indicate that its representation better captures precise local geometry together with coherent global CAD structure.

## 5. Conclusions

This paper presents Brep2Shape, a novel self-supervised pre-training method that aligns the fundamental gap between abstract boundary representations and intuitive shape representations. The alignment is achieved by a Dual Transformer backbone, which leverages topology attention to maintain structural consistency between face and edge streams. Extensive experiments demonstrate Brep2Shape yields generalizable representations through pre-training. When fine-tuned on downstream tasks, it remarkably outperforms state-of-the-art models in both accuracy and convergence speed.

## Impact Statement

This paper introduces a novel self-supervised framework for boundary representation (B-rep), advancing the intersection of geometric modeling and representation learning. By bridging the gap between abstract parametric entities and intuitive shape geometry, our work has the potential to significantly reduce the dependency on large-scale labeled datasets in industrial design, thereby lowering the barrier for AI adoption in engineering and manufacturing. While this advancement contributes to the broader goal of intelligent CAD systems, we do not foresee any specific negative societal consequences that require immediate highlighting.

## Acknowledgements

This work was supported by Beijing Scholar Program and National Engineering Research Center for Big Data Software.

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

# A. Implementation Details

## A.1. Datasets

**Pre-training dataset** Large-scale and diverse corpora are crucial for learning robust representations of B-rep solids, as real-world CAD models vary substantially in topology, local geometry, and modeling styles. To support our pre-training objective, we curate *Brep2Shape-250k*, a corpus of 250,000 B-rep solids compiled from multiple public CAD datasets, covering both mechanical parts and shapes derived from modeling operations. *Brep2Shape-250k* aggregates models from eight sources, including MFCAD (Cao et al., 2020), MFCAD++ (Colligan et al., 2022), TM-CAD (Zou & Zhu, 2025), FabWave (Angrish et al., 2019), Fusion360Seg (Lambourne et al., 2021), Fusion360Ass (Willis et al., 2022), Fusion360Rec (Willis et al., 2021), and DeepCAD (Wu et al., 2021). For

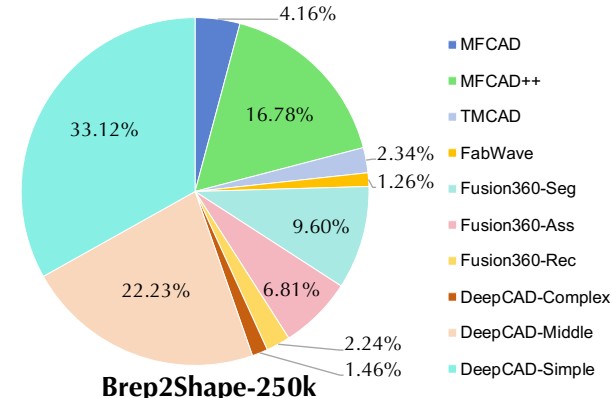

*Figure 8.* Ratios of data sources in *Brep2Shape-250k*, the pre-training corpora of Brep2Shape. Detailed statistics are provide in Table 9.

DeepCAD, we further group models into three complexity levels based on the number of faces, using 10, 30, and 60 as thresholds. Figure 8 visualizes the source distribution, and Table 9 reports the exact counts and ratios. The construction of this dataset required substantial engineering effort, involving *tens of days of continuous processing on 30 CPU cores* to curate, clean, and standardize the data.

*Table 9.* Key statistics of *Brep2Shape-250k*, the pre-training dataset of Brep2Shape.

| Source | MFCAD | MFCAD++ | TMCAD | FabWave | Fusion360Seg | Fusion360Ass | Fusion360Rec | DeepCAD-Complex | DeepCAD-Middle | DeepCAD-Simple | Total |
|---|---|---|---|---|---|---|---|---|---|---|---|
| #Models | 10811 | 43587 | 6066 | 3270 | 24924 | 18145 | 5813 | 3548 | 53754 | 80082 | 250000 |
| Ratio | 4.16% | 16.78% | 2.34% | 1.26% | 9.60% | 6.81% | 2.24% | 1.46% | 22.23% | 33.12% | 100% |

**Downstream benchmarks** We extensively evaluate our model on four benchmarks. Note that these benchmarks involve classification and segmentation tasks for solid mode modeling.

**FabWave** FabWave (Angrish et al., 2019) focuses on fine-grained classification of engineering parts. It contains 4,572 B-rep models across 45 categories, covering common parts such as brackets, gears, and O-rings. The input is a B-rep model and the output is a category label. The average numbers of edges and faces per model are $\overline{N_e} = 110.70$ and $\overline{N_f} = 22.10$, respectively.

**TMCAD** TMCAD (Zou & Zhu, 2025) comprises more than 10,000 real-world mechanical design models collected from the Internet. 7599 processed and valid models are retained for classification, including 10 classes of standard parts such as bearings, couplings, and flanges. The average numbers of edges and faces per model are $\overline{N_e} = 400.36$ and $\overline{N_f} = 75.44$, respectively.

**MFCAD++** MFCAD++ (Colligan et al., 2022) is designed for machining feature recognition, treating the problem as a face segmentation task. The dataset comprises 59,665 B-rep models. The goal is to identify geometric features such as passages, pockets, and slots, assigning one of 25 semantic labels to each face of the model. The average numbers of edges and faces per model are $\overline{N_e} = 157.36$ and $\overline{N_f} = 29.91$, respectively.

**Fusion360Seg** Fusion360Seg (Lambourne et al., 2021) facilitates B-rep semantic segmentation using data from the Fusion 360 Gallery Dataset. It includes 35,858 models. The objective is to label each face according to the modeling operation source, with 8 categories including Extrude Side, Extrude End, and Chamfer. The average numbers of edges and faces per model are $\overline{N_e} = 69.59$ and $\overline{N_f} = 14.70$, respectively.

## A.2. Metrics

**Shape reconstruction loss**  In the pre-training phase, the model predicts the shape representation $\hat{\mathcal{U}} = f_\theta(\mathcal{P}, \mathcal{G})$ and $\hat{\mathcal{U}} = \{\hat{\mathcal{U}}_{f_j}\}_{j=1}^{N_f} \cup \{\hat{\mathcal{U}}_{e_k}\}_{k=1}^{N_e}$ where each entity target is represented as a set of sampled 3D points. Specifically, for the $j$-th face and the $k$-th edge, the targets are $\mathcal{U}_{f_j} \in \mathbb{R}^{m \times 3}$ and $\mathcal{U}_{e_k} \in \mathbb{R}^{m \times 3}$, respectively, where padding is applied to ensure a uniform size $m$. Let $\Omega_{f_j}$ and $\Omega_{e_k}$ denote the sets of valid (non-padded) point indices for faces and edges, and $\mathcal{U}_x[p]$ ($x$ for $f_j$ or $e_k$) denotes the 3D points at index $p$. We minimize the mean squared error (MSE) over valid points only. The overall pre-training loss is defined as:

$$\mathcal{L}_{\text{pre}} = \mathcal{L}_{\text{face}} + \mathcal{L}_{\text{edge}}. \tag{3}$$

$$\mathcal{L}_{\text{face}} = \frac{1}{N_f} \sum_{j=1}^{N_f} \frac{1}{|\Omega_{f_j}|} \sum_{p \in \Omega_{f_j}} \left\| \hat{\mathcal{U}}_{f_j}[p] - \mathcal{U}_{f_j}[p] \right\|_2^2, \tag{4}$$

$$\mathcal{L}_{\text{edge}} = \frac{1}{N_e} \sum_{k=1}^{N_e} \frac{1}{|\Omega_{e_k}|} \sum_{p \in \Omega_{e_k}} \left\| \hat{\mathcal{U}}_{e_k}[p] - \mathcal{U}_{e_k}[p] \right\|_2^2, \tag{5}$$

**Accuracy (Acc)**  We evaluate the performance of both classification and segmentation tasks using Accuracy. It measures the proportion of correctly predicted samples (or faces in segmentation) out of the total number of samples $N$:

$$\text{Acc} = \frac{1}{N} \sum_{i=1}^{N} \mathbb{I}(\hat{y}_i = y_i) \tag{6}$$

where $\hat{y}_i$ is the predicted label and $y_i$ is the ground truth label. $\mathbb{I}(\cdot)$ is the indicator function.

**Intersection over Union (IoU)**  For the segmentation task, we also employ the Intersection over Union (IoU) metric to assess the overlap between the predicted and ground truth regions for each class. The IoU for a specific class $c$ is defined as:

$$\text{IoU}_c = \frac{\text{Prediction}_c \cap \text{GroundTruth}_c}{\text{Prediction}_c \cup \text{GroundTruth}_c} \tag{7}$$

and we report the mean IoU across all $C$ classes: $\text{IoU} = \frac{1}{C} \sum_{c=1}^{C} \text{IoU}_c$.

## A.3. Implementations

Our models are implemented using the PyTorch framework (Paszke et al., 2019) and trained on an NVIDIA A100 GPU. The optimization process utilizes the AdamW optimizer (Loshchilov & Hutter, 2019) with a cosine annealing learning rate schedule. For Brep2Shape, the initial learning rate is set to $1 \times 10^{-4}$ with a weight decay of 0.01 for pre-training and $2 \times 10^{-4}$ with the same weight decay for fine-tuning. Unless otherwise specified, the edge and face tokenizers each use a 3-layer Transformer with 4 attention heads and a hidden dimension of 128, while the Dual Transformer consists of 6 Transformer layers with 4 attention heads and a hidden dimension of 128. Detailed hyperparameter configurations for all models are summarized in Table 10.

*Table 10.* Hyperparameter settings and configurations for different models. Transformer-based models (e.g., Brep2Shape and BRT) specify the number of attention heads, while UVNet and AAGNet do not use attention mechanisms and therefore have no head configuration (denoted by "–"). * For Brep2Shape, the batch size is set to 16 during pre-training and 32 during fine-tuning. This is default config for Brep2Shape.

| Model | Edge Tokenizer | | | Face Tokenizer | | | Dual Transformer | | | Batch Size | Epochs |
|---|---|---|---|---|---|---|---|---|---|---|---|
| | Layers | Heads | Hidden Dim. | Layers | Heads | Hidden Dim. | Layers | Heads | Hidden Dim. | | |
| Brep2Shape | 3 | 4 | 128 | 3 | 4 | 128 | 6 | 4 | 128 | 16* | 100 |
| BRT | 2 | 4 | 64 | 2 | 4 | 64 | 2 | 8 | 64 | 16 | 350 |
| UVNet | 3 | – | 64 | 3 | – | 64 | 2 | – | 128 | 128 | 350 |
| AAGNet | 1 | – | 64 | 1 | – | 64 | 3 | – | 128 | 256 | 350 |

**BRT**  BRT (Zou & Zhu, 2025) adopts a hierarchical Transformer architecture consisting of 6 Transformer encoder layers, with 2 layers each for the edge, face, and global encoders. The embedding dimension is set to 64, and the feed-forward network (FFN) uses a hidden dimension of 512. A dropout rate of 0.25 is applied throughout the network.

**UV-Net**  UV-Net (Jayaraman et al., 2021) The curve encoder consists of 3 1D convolutional layers with hidden dimensions of 64, 128, and 256, followed by adaptive average pooling and a fully connected layer outputting a 64-dimensional embedding. Similarly, the surface encoder comprises 3 2D convolutional layers with an identical channel configuration.

**AAGNet**  AAGNet (Wu et al., 2024a) utilizes linear networks to encode B-rep face and edge attributes (64-dim), followed by layer normalization. It includes a CNN-based encoder for node grid features. The core graph encoder processes these node and edge embeddings using 3 GNN layers. For classification, we adapt the original segmentation design by extracting global graph features and passing them through a 2-layer MLP head with a hidden dimension of 64 and layer normalization.

## B. Additional Analysis

### B.1. Sample Density Sensitivity

For each B-rep entity, we obtain a spatial point-based shape representation by discretely sampling its parametric domain. The sampling density is controlled by the number of sampled points $m$ per primitive. In our default setting, Brep2Shape predicts $m = 3$ points for each primitive. To evaluate the sensitivity of the model to sampling density, we conduct experiments with $m = 1$ and $m = 5$. All models are pre-trained on Brep2Shape-25K using a 6-layer Dual Transformer. The results are reported in Table 11. As $m$ increases, the pre-training loss rises, indicating higher prediction difficulty due to the increased output dimensionality. However, downstream performance on TMCAD consistently improves with larger $m$. This suggests that denser sampling enables the model to capture and localize geometric details more effectively. We therefore adopt $m = 3$ as a balanced choice, providing sufficient geometric interpretability while maintaining stable training and strong downstream performance.

*Table 11.* Sensitivity analysis with different sampling densities $m$. The default setting for all experiments ($m = 3$) is highlighted in **bold**.

| Sampling Density | TMCAD | Pre-train Loss |
|---|---|---|
| $m = 1$ | 0.8125 | 1.97 |
| $\mathbf{m = 3}$ | **0.8212** | **3.226** |
| $m = 5$ | 0.8247 | 3.793 |

### B.2. Full Results for Data Efficiency Analysis

Table 12 reports the full numerical results of the data efficiency analysis corresponding to Figure 5, providing exact accuracy values on MFCAD++ and TMCAD across different labeling regimes.

*Table 12.* Data efficiency analysis on MFCAD++ and TMCAD under varying numbers of labeled training samples.

| Method | MFCAD++ | | | | | TMCAD | | | | |
|---|---|---|---|---|---|---|---|---|---|---|
| | 100 | 1k | 10k | 20k | 47k(full) | 100 | 1k | 2k | 4k | 6.4k(full) |
| UV-Net | 0.4381 | 0.8559 | 0.9708 | 0.9848 | 0.9902 | 0.4147 | 0.6761 | 0.7301 | 0.7670 | 0.7787 |
| BRT | 0.5017 | 0.8520 | 0.9293 | 0.9357 | 0.9573 | 0.4479 | 0.7153 | 0.7552 | 0.7760 | 0.8090 |
| Brep2Shape | **0.5224** | **0.8936** | **0.9873** | **0.9897** | **0.9934** | **0.4792** | **0.7361** | **0.7760** | **0.8026** | **0.8264** |

### B.3. Full Results for Pre-training Strategy Ablations

Table 13 reports the full results of different pre-training strategies. In addition to the pre-training losses, we evaluate the resulting representations on downstream tasks to examine whether lower reconstruction loss translates into better transferability. Pre-training only the tokenizers removes the Dual Transformer during pre-training, while the variant without edge-level supervision removes the edge prediction loss. Both components contribute to downstream performance: removing the Dual Transformer leads to a clear drop on TMCAD, and removing edge-level supervision also degrades transfer performance. These results confirm that both global topological encoding and edge-level spatial alignment are important for learning generalizable B-rep representations.

*Table 13.* Full results for pre-training strategy ablations. Face Loss and Edge Loss are reported in units of $10^{-3}$. TMCAD reports classification accuracy, while MFCAD++ reports face segmentation Acc./IoU.

| Method | Face Loss | Edge Loss | TMCAD | MFCAD++ |
|---|---|---|---|---|
| Pre-train Tokenizers Only | 2.142 | 1.287 | 80.40 | 99.16 / 97.56 |
| Dual Trm. w/o Edge-Level Sup. | 1.872 | – | 81.43 | 99.18 / 97.56 |
| Brep2Shape | **1.855** | **1.238** | **82.64** | **99.33 / 98.02** |

## B.4. Statistical Analysis

Table 14 reports fine-tuning results of Brep2Shape under three different random seeds. Overall, performance is highly stable across runs on both classification and segmentation benchmarks. In particular, FabWave and MFCAD++ exhibit negligible variance, with classification Acc and segmentation Acc/IoU remaining nearly identical across seeds. The largest fluctuation is observed on Fusion360Seg, where IoU varies slightly across runs, indicating that this dataset is comparatively more sensitive to optimization randomness. Despite this, the overall trends and competitive performance remain consistent, suggesting that Brep2Shape fine-tuning is robust to seed variation.

*Table 14.* Performance of Brep2Shape across three random seeds. We report Acc for classification and Acc/IoU for segmentation.

| Seed | FabWave | TMCAD | MFCAD++ | | Fusion360Seg | |
|---|---|---|---|---|---|---|
| | | | Acc | IoU | Acc | IoU |
| Seed 1 | 0.9924 | 0.8264 | 0.9934 | 0.9802 | 0.9690 | 0.8381 |
| Seed 2 | 0.9999 | 0.8299 | 0.9931 | 0.9798 | 0.9691 | 0.8346 |
| Seed 3 | 0.9999 | 0.8229 | 0.9934 | 0.9806 | 0.9629 | 0.8249 |
| Mean | 0.9974 | 0.8264 | 0.9933 | 0.9802 | 0.9670 | 0.8325 |
| Standard Deviation | 0.0043 | 0.0035 | 0.0002 | 0.0004 | 0.0036 | 0.0068 |

## B.5. Computation Costs

All experiments are conducted on an NVIDIA A100 GPU. As summarized in Table 15, the pre-training stage consumes 38 GPU-hours, while fine-tuning on downstream datasets requires 17.7 GPU-hours on MFCAD++, 6.8 GPU-hours on Fusion360Seg, 1.0 GPU-hour on FabWave, and 2.2 GPU-hours on TMCAD.

*Table 15.* GPU-hour statistics for pre-training and fine-tuning across different datasets.

| Stage | Pre-training (250k) | MFCAD++ | Fusion360Seg | FabWave | TMCAD |
|---|---|---|---|---|---|
| GPU-hours (A100 × h) | 38 | 17.7 | 6.8 | 1.0 | 2.2 |

## C. Geometric Foundations for B-reps

In this section, we provide more details about Bézier, B-splines and NURBS(Piegl & Tiller, 2012).

**Bézier**  Bézier curves are the most fundamental parametric curves, defined by a set of control points $\mathbf{P}_i$. A degree-$n$ Bézier curve is expressed as a linear combination of Bernstein basis polynomials:

$$\mathbf{C}(u) = \sum_{i=0}^{n} B_{i,n}(u)\mathbf{P}_i, \quad u \in [0,1] \tag{8}$$

where the Bernstein basis polynomial is $B_{i,n}(u) = \binom{n}{i}u^i(1-u)^{n-i}$. Bézier surfaces extend Bézier curves to two parameters $(u,v)$ with a control net $\{\mathbf{P}_{i,j}\}$. A tensor-product Bézier surface of bi-degree $(n,m)$ is defined as

$$\mathbf{S}(u,v) = \sum_{i=0}^{n} \sum_{j=0}^{m} B_{i,n}(u)\, B_{j,m}(v)\, \mathbf{P}_{i,j}, \quad (u,v) \in [0,1]^2. \tag{9}$$

However, the basis functions have global support over $[0,1]$, so perturbing a single control point generally affects the entire entitiy. Moreover, being polynomial, they cannot exactly represent conic sections (e.g., circles, ellipses) in Euclidean space.

**B-Splines**    B-splines generalize Bézier representations by introducing a knot vector $U = \{u_0, \ldots, u_m\}$ and the associated basis functions $N_{i,p}(u)$ of degree $p$. Given degree $p$, control points $\{\mathbf{P}_i\}_{i=0}^n$, and $m = n + p + 1$, a B-spline curve is

$$\mathbf{C}(u) = \sum_{i=0}^n N_{i,p}(u)\,\mathbf{P}_i. \tag{10}$$

The basis functions are defined by the Cox–de Boor recursion:

$$N_{i,0}(u) = \begin{cases} 1, & u_i \le u < u_{i+1}, \\ 0, & \text{otherwise}, \end{cases} \tag{11}$$

$$N_{i,p}(u) = \frac{u - u_i}{u_{i+p} - u_i} N_{i,p-1}(u) + \frac{u_{i+p+1} - u}{u_{i+p+1} - u_{i+1}} N_{i+1,p-1}(u). \tag{12}$$

A degree-$p$ Bézier curve is a special case of a B-spline curve with no internal knots and end knots of multiplicity $(p+1)$. Modifying a control point affects the curve only within the support of the corresponding basis functions, making B-splines highly suitable for modeling complex shapes. B-spline surfaces are obtained by extending the above basis construction to two parameters via a tensor-product of basis functions defined on two knot vectors. Consequently, they inherit the same locality and flexibility: modifying a control point in the control net influences only a local region of the surface in parameter space. A single Bézier surface patch can be viewed as a special case of a B-spline surface whose knot vectors contain no internal knots and use end knots with multiplicity $(p+1)$ and $(q+1)$, respectively.

**NURBS**    NURBS further extend B-splines by introducing weights $w_i$, enabling exact representation of conic sections (e.g., circles and ellipses) and related surfaces (e.g., cylinders). A B-spline is a special case of NURBS with $w_i = 1$ for all $i$. Given degree $p$, knot vector $U$, control points $\{\mathbf{P}_i\}_{i=0}^n$, and weights $\{w_i\}_{i=0}^n$, a NURBS curve is

$$\mathbf{C}(u) = \frac{\sum_{i=0}^n N_{i,p}(u)\,w_i\,\mathbf{P}_i}{\sum_{i=0}^n N_{i,p}(u)\,w_i}. \tag{13}$$

To linearize geometric operations, lift control points to homogeneous coordinates $\mathbf{P}_i^w = (w_i x_i,\, w_i y_i,\, w_i z_i,\, w_i) \in \mathbb{R}^4$. In homogeneous space, many NURBS operations reduce to standard B-spline operations on $\{\mathbf{P}_i^w\}$. In this space, NURBS operations degenerate into standard B-Spline operations, greatly simplifying algorithmic processing.

## D. Details on Bézier Decomposition

To bridge the gap between complex NURBS geometric entities in B-rep models and the structured input requirements of transformer models (i.e., fixed-length sequences of control points), we follow the approach of BRT to perform a standardized decomposition for both edges and faces (Zou & Zhu, 2025). Moreover, we analyze why this decomposition can be near-lossless.

### D.1. Decomposition Methods

Building upon the homogeneous representation, we now detail the decomposition process for edges and faces. This standardization step is crucial for converting variable-length NURBS data to fixed-topology tensor inputs.

**Edge Decomposition**    Each edge in the B-rep model is represented as a degree-$p$ NURBS curve with knot vector $U$. Since the length of the knot vector varies, we employ *Boehm's knot insertion algorithm* (Boehm, 1980) to decompose it into independent Bézier segments. Specifically, for each internal knot $u_k$ with multiplicity $s$, if $s < p$, we insert $u_k$ for $(p - s)$ times until its multiplicity reaches $p$. Knot insertion changes the basis functions in the parameter domain; to keep the curve shape $\mathbf{C}(u)$ invariant, the (homogeneous) control points must be updated accordingly. Inserting a knot $\hat{u} \in [u_k, u_{k+1}]$ yields a refined set of homogeneous control points $\{\mathbf{Q}_i^w\}$ computed by linear interpolation:

$$\mathbf{Q}_i^w = (1 - \alpha_i)\mathbf{P}_{i-1}^w + \alpha_i \mathbf{P}_i^w, \quad \alpha_i = \frac{\hat{u} - u_i}{u_{i+p} - u_i}. \tag{14}$$

After recursively inserting knots, the refined B-spline can be segmented into multiple Bézier curves, where each segment is determined by exactly $(p+1)$ control points, producing fixed-length tensor inputs for neural network processing.

**Face Decomposition** For face entities, the geometry is a tensor-product NURBS surface $\mathbf{S}(u, v)$. We apply the following three-stage pipeline to unify the primitive representation.

**Step 1: Generation of Bézier Rectangles** Similarly to edge decomposition, we apply knot insertion independently along both the $u$ and $v$ directions until every *internal* knot reaches multiplicity $p$ and $q$, respectively (degrees along $u$ and $v$). This decomposes the surface into a grid of tensor-product Bézier patches in homogeneous space. For each patch, the homogeneous surface can be written as

$$\mathbf{S}^w_{\text{rect}}(u, v) = \sum_{i=0}^{p} \sum_{j=0}^{q} B_i^p(u) \, B_j^q(v) \, \mathbf{P}^w_{i,j}, \qquad (u, v) \in [0, 1]^2, \tag{15}$$

where $B_i^p(\cdot)$ and $B_j^q(\cdot)$ are univariate Bernstein basis polynomials. Here we place the degree as a superscript for convenience in the following explanation.

**Step 2: Transformation to Bézier Triangles** To unify the input format for the neural network, each tensor-product Bézier patch over $[0, 1]^2$ is split along the diagonal $u + v = 1$ into two triangular patches (Farin, 1986). For the *lower* triangle $\Delta = \{(u, v) \mid u \geq 0, \ v \geq 0, \ u + v \leq 1\}$, define the barycentric coordinates $t = 1 - u - v$. A triangular Bézier surface of total degree $d$ is written as

$$\mathbf{T}^w(u, v) = \sum_{i \geq 0, \ j \geq 0, \ i+j \leq d} B^d_{i,j}(u, v) \, \mathbf{V}^w_{i,j}, \tag{16}$$

where the bivariate Bernstein basis functions are defined as:

$$B^d_{i,j}(u, v) = \frac{d!}{i! \, j! \, (d - i - j)!} \, u^i v^j t^{d-i-j}. \tag{17}$$

Since a tensor-product polynomial of bi-degree $(p, q)$ contains monomials up to total degree $p + q$, we set $d = p + q$ to capture the full geometry without information loss.

By expanding both polynomial bases and equating their coefficients, we derive a linear mapping that computes the new control points $\mathbf{V}^w$ directly from the original rectangular points $\mathbf{P}^w$. The derivation proceeds as follows: Substituting the identity $1 - u = v + t$ and $1 - v = u + t$ (where $t = 1 - u - v$) into the tensor-product Bernstein basis functions:

$$B_a^p(u) \, B_b^q(v) = \binom{p}{a} \binom{q}{b} u^a (v + t)^{p-a} \, v^b (u + t)^{q-b} \tag{18}$$

then expand the binomials and collect terms with total degree $d = p + q$. This yields the explicit conversion formula

$$\mathbf{V}^w_{i,j} = \sum_{a=0}^{p} \sum_{b=0}^{q} \left[ \binom{p}{a} \binom{q}{b} \binom{p-a}{j-b} \binom{q-b}{i-a} \frac{i! \, j! \, (d - i - j)!}{d!} \right] \mathbf{P}^w_{a,b}, \quad d = p + q, \tag{19}$$

where $\binom{n}{k} = 0$ if $k < 0$ or $k > n$.

### Step 3: Quadtree Boundary Decomposition

For trimmed surfaces, the valid geometry domain is bounded by curves defined in parametric space $(u, v)$. To accurately capture irregular boundaries while maintaining efficiency, we employ an adaptive quadtree subdivision approach. We recursively subdivide the $(u, v)$ domain and classify each rectangle based on its relationship to the trim boundary: interior rectangles are retained without subdivision, exterior rectangles are discarded, and boundary-crossing rectangles are further subdivided in parametric space until reaching the maximum depth or sufficient local refinement (e.g., chord-to-arc ratio). All resulting valid leaf rectangles are converted into triangular patches following Step 2. This produces a unified collection of triangular Bézier patches with adaptive resolution: coarse triangulation in smooth interior regions and fine triangulation near complex boundaries, optimally balancing geometric accuracy and computational efficiency.

### D.2. Why the Decomposition Is Near-lossless?

While the decomposition is mathematically exact for the interior of complete surfaces, handling trimmed boundaries necessitates the subdivision and approximation of cells intersecting the trimming curves. Consequently, the primary source

of geometric deviation in our representation stems from this boundary discretization. In this section, we analyze the error induced by the quadtree boundary decomposition. Specifically, we evaluate the fidelity of our piecewise linear approximation and demonstrate that as the discretization is refined, the global approximation error vanishes at a quadratic rate.

This result follows directly from standard interpolation error analysis in numerical methods, and we simply adapt the classical proof strategy to our boundary discretization setting (Gautschi, 2011; Ascher & Greif, 2011). This theoretical guarantee ensures that the discrete representation remains near-lossless for our experiments.

**Problem Setup**  Let $\mathcal{C}(t) : [a, b] \to \Omega \subseteq \mathbb{R}^2$ be a regular $C^2$ continuous curve representing a trimming boundary in the parametric domain. The *quadtree boundary decomposition* adaptively partitions the parameter interval $[a, b]$ into $N$ small segments $\{[t_i, t_{i+1}]\}_{i=1}^N$. To formalize the error analysis, we denote the parametric step size of each subdivided segment as $h_i = t_{i+1} - t_i$, with $h = \max_i\{h_i\}$ representing the *maximum step size*. Within each parametric interval $[t_i, t_{i+1}]$, the continuous trim curve $\mathcal{C}(t)$ is approximated by the first-order Lagrange interpolant $\mathcal{L}_i(t)$. Given the total arc length of the boundary curve $L = \int_a^b \|\mathcal{C}'(t)\| dt$, we can evaluate the fidelity of this approximation through the root mean square error (RMSE), defined as: $E_{RMSE} = \sqrt{\frac{1}{L} \int_a^b \|\mathcal{C}(t) - \mathcal{L}(t)\|^2 dt}$.

**Theorem D.1** (**Quadratic convergence for boundary approximation**). *For a $C^2$ continuous boundary curve $\mathcal{C}$ discretized via the first-order Lagrange interpolant $\mathcal{L}_i(t)$ with a maximum step size $h$, the RMSE $E_{RMSE}$ satisfies:*

$$E_{RMSE} \leq O(h^2) \quad as \quad h \to 0 \tag{20}$$

*This indicates that the approximation error vanishes quadratically with respect to the maximum step size.*

*Proof.*  The proof proceeds by analyzing the error propagation from local interpolants to the global deviation.

According to the error formula for first-order Lagrange interpolation of a $C^2$ continuous function, the pointwise Euclidean distance between the curve $\mathcal{C}(t)$ and the linear chord $\mathcal{L}_i(t)$ on any interval $[t_i, t_{i+1}]$ is bounded by (Gautschi, 2011),

$$\|\mathcal{C}(t) - \mathcal{L}_i(t)\| \leq \frac{h_i^2}{8} \sup_{\xi \in [t_i, t_{i+1}]} \|\mathcal{C}''(\xi)\| \leq O(h^2) \tag{21}$$

Therefore, the squared error contribution over a single segment is obtained by integrating the squared pointwise deviation,

$$\int_{t_i}^{t_{i+1}} \|\mathcal{C}(t) - \mathcal{L}_i(t)\|^2 dt \leq \int_{t_i}^{t_{i+1}} (O(h^2))^2 dt = \int_{t_i}^{t_{i+1}} O(h^4) dt = O(h^5) \tag{22}$$

Assuming the segments are quasi-uniform, the total number of segments $N$ is proportional to $L/h$. We sum the contributions across all $N$ segments and apply the definition of $E_{RMSE}$,

$$E_{RMSE}^2 = \frac{1}{L} \sum_{i=1}^N \int_{t_i}^{t_{i+1}} \|\mathcal{C}(t) - \mathcal{L}_i(t)\|^2 dt \leq \frac{1}{L} \cdot \left( \frac{L}{h} \cdot O(h^5) \right) = O(h^4) \tag{23}$$

Taking the square root of the normalized sum yields,

$$E_{RMSE} \leq \sqrt{O(h^4)} = O(h^2) \tag{24}$$

This confirms that the RMSE converges to zero at a *quadratic rate* with respect to maximum step size $h$. □

**Practical Implementation**  In practical implementations, employing a fixed parametric step size $h$ as the refinement criterion presents significant limitations. Primarily, $h$ is an absolute quantity that lacks *scale-invariance*. For instance, a fixed threshold may lead to insufficient geometric fidelity for micro-scale features while causing computational redundancy in large-scale components.

To address this, we introduce an adaptive control mechanism based on a dimensionless constant, *chord-to-arc ratio* $\tau$. It dynamically modulates the subdivision frequency according to the local curvature of the curve. By applying a standard

Taylor expansion to a small curve segment with respect to the parametric step $h$, the relationship between the chord length and the arc length can be derived through their respective power series. For a curve with local curvature $\kappa$, the chord length is $h - \frac{1}{24}\kappa^2 h^3 + O(h^5)$ and the arc length is $h + O(h^3)$ (Piegl & Tiller, 2012),

$$\tau = \frac{\text{Chord}(h)}{\text{Arc}(h)} = 1 - \frac{\kappa^2 h^2}{24} + O(h^4). \tag{25}$$

This formulation reveals that the deviation $(1 - \tau)$ is directly proportional to $\kappa^2 h^2$. Substituting this relationship into our convergence theorem, it follows that the error bound is effectively governed by $E_{RMSE} \leq C \cdot (1 - \tau)$. Such a control logic not only aligns with the theoretical quadratic convergence of the discretization but also provides a scale-independent and *error-controllable* metric, ensuring that *Brep2Shape* captures complex industrial geometries with near-lossless precision.

Overall, the above inequality demonstrates that by prescribing the threshold $\tau$ sufficiently close to 1, the global approximation error can be sufficiently small. This confirms that our discretization scheme is *error-controllable*, theoretically allowing the approximation to approach the original geometry with arbitrary precision. However, in practice, a finer discretization (higher $\tau$) imposes a heavier computational burden on both pre-processing and subsequent learning. To achieve a balance between geometric fidelity and computational efficiency, we empirically set $\tau = 0.995$. This configuration effectively maintains high-precision boundary representation while ensuring rapid convergence and inference performance in our experiments.

## E. Additional Visualizations

In this section, we provide additional qualitative results as a supplement to Figure 6. Specifically, we show more segmentation examples on Fusion360Seg in Figure 9 and more classification examples on TMCAD in Figure 10.

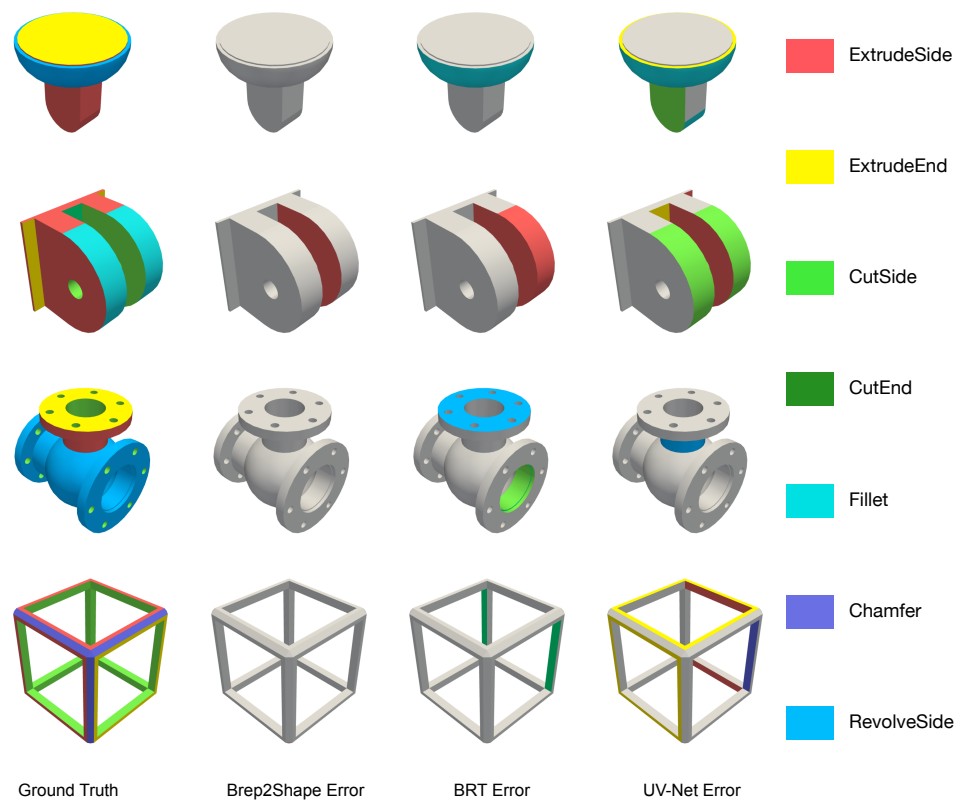

*Figure 9.* Additional visualizations for segmentation on Fusion360Seg.

## F. Limitations and Future Work

Despite its effectiveness, Brep2Shape has some limitations that provide avenues for future research.

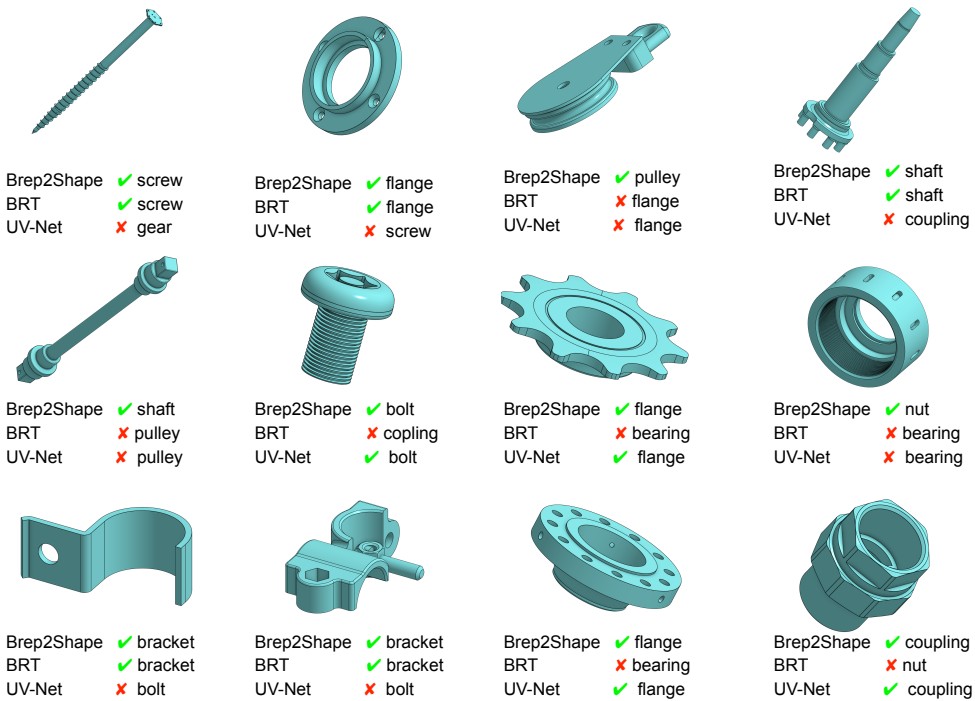

*Figure 10.* Additional visualizations for classification on TMCAD.

**(i)** Brep2Shape balances mathematical continuity and model compatibility via standardized Bézier decomposition, satisfying most industrial CAD requirements. However, for *ultra-precision applications* (e.g., aerospace manufacturing), fixed-order decomposition can introduce certain approximation biases. These residuals are negligible for representative learning but provide a focus for future optimization in these precision-critical domains.

**(ii)** While we demonstrate state-of-the-art results on several benchmarks, the diversity of downstream tasks *in this domain* remains somewhat constrained. Due to the lack of established high-quality datasets for edge-level tasks (e.g., edge classification), our current evaluation primarily focuses on face-level and shape-level understanding. Enriching the domain with more diverse and comprehensive tasks would be a valuable next step.

**(iii)** Industrial CAD designs often consist of complex assemblies. Extending Brep2Shape to handle multi-body interdependencies and joint constraints is a primary focus of our future work, aiming toward a truly holistic understanding of large-scale engineering systems.

