# OpenReview forum: "Brep2Shape: Boundary and Shape Representation Alignment via Self-supervised Transformers"
_ICML.cc/2026/Conference — ICML 2026 regular_

### Official Review · Reviewer_Btop · 2026-03-04

**Soundness:** 3
**Presentation:** 3
**Significance:** 3
**Originality:** 3
**Overall Recommendation:** 4
**Confidence:** 2

**Summary:**

This paper addresses the learning challenge of boundary representation (B-rep) in the field of computer-aided design (CAD) and proposes a self-supervised pre-training method named Brep2Shape. The core idea is to establish the alignment between the abstract boundary representations with intuitive shape representations.

**Compliance With Llm Reviewing Policy:**

Affirmed.

**Final Justification:**

I appreciate the authors' effort in addressing my questions. I keep the rating as weak acceptance.

**Key Questions For Authors:**

Questions:

1. When performing uniform sampling in the parametric domain, for surfaces with severe parametric distortion (Distorted surfaces), the distribution of sampling points in 3D space may be extremely uneven. How does the model handle the learning bias caused by the inconsistent sampling density?

2. When Surface and Edge streams are encoded in parallel, does the problem of feature homogenization exist? Is topological attention bias sufficient to distinguish the two completely different fluid geometric properties?

3. Experiments show that the downstream task performs better when the sampling point density is low, but the model ultimately chose this. Will the increase in more complex industrial parts lead to diminishing marginal effects?

4. In the Dual Transformer, how are the information exchange frequencies of line branches and surface branches determined? Have you considered a deeper fusion strategy across modalities (from surface to line)?

5. Currently, the evaluation mainly focuses on surface classification and segmentation. Can this pre-trained model be directly used for completion tasks (such as automatically completing missing CAD surfaces)?

**Limitations:**

yes

**Strengths And Weaknesses:**

Strengths:

1. Unlike UV-Net, Brep2Shape directly Bridges the "Representation Gap" between continuous parameter representation and discrete space intuition through geometric reconstruction tasks.

2. This paper studies the lossless decomposition from NURBS to Bezier and provides a mathematical proof of the convergence rate of boundary approximation errors, enhancing the theoretical persuasiveness of the method.

3. In multiple downstream benchmark tests such as MFCAD++ and Fusion360Seg, Brep2Shape has set new SOTA records in both accuracy and convergence speed. Especially in the aspect of Limited Data Generalization, it performs astonishingly.

Weaknesses:

1. Currently, models mainly focus on the understanding of Single bodies. For complex assemblies that are common in the industry and contain multiple components with mutual motion constraints, their modeling capabilities remain to be proven.

2. The paper mentions that Edge-level supervision is used to ensure spatial consistency between surfaces. However, in the current experiments, edges are only regarded as "anchor points" for auxiliary alignment, lacking in-depth semantic mining for edge features (such as hard edges vs. cut edges)

3. The experiments mainly focus on classification and segmentation tasks. As a self-supervised pre-trained model, the lack of verification for more challenging downstream tasks such as "Geometry Completion" or "Parameter Regression" weakens the proof of its "general representation" ability.

4. Although the paper introduces attention bias based on surface adjacency graphs, when dealing with industrial parts with an extremely large number of fine Fillets or complex array features, this topological bias may cause the model to overly focus on local connections while ignoring long-range geometric symmetries or functional associations.

5. Brep2Shape mainly achieves alignment through the reconstruction "from parameters to spatial points." However, this one-way mapping may overlook a key inverse conundrum in B-rep learning: how to reverse-constrain the rigor of the parameter space from shape features with semantic awareness. The current task design focuses more on enabling the model to "understand" geometry rather than "comprehend" the parameter logic that generates these geometries.

---

> ### Author Rebuttal · Authors · 2026-03-30
>
> We sincerely thank Reviewer Btop for providing insightful suggestions.
>
> > W1. Extension to assemblies
>
> We acknowledge that Brep2Shape focuses on single-body understanding. Assembly-level reasoning could be approached by a hierarchical graph where bodies serve as nodes and inter-body relations as edges, with cross-body attention layers on top of the Dual Trm. We note this limitation in App.F (iii) and plan to explore it in future work.
>
> > W2. Lack of semantic mining for edge
>
> Edge-level supervision serves a dual purpose: enforcing spatial consistency at shared boundaries to improve surface learning and producing meaningful edge representations. As shown in Q2, pre-trained features are clearly distinguishable, suggesting semantically meaningful edge learning. Explicit edge-level evaluation remains unexplored due to absent benchmarks, which we hope our work can help motivate.
>
> > W3 & Q5. Lack of evaluation on generative tasks
>
> We clarify that our claim is **generalizable**, not **general** representations: Brep2Shape generalizes across understanding tasks, not claiming universality. This reflects a principle—the pre-training objective shapes downstream capabilities. Just as MAE excels at recognition but requires adaptation for generation, our reconstruction-based pre-training naturally aligns with understanding tasks. Extending to generative tasks would require complementary objectives, an exciting future direction.
>
> > W4. Topology bias may cause over-focus on local connections
>
> The topology attention is an additive bias, not a mask—self-attention still computes pairwise interactions across all tokens. Modern Transformers handle 128K+ token sequences even with local attention, such as DeepSeek Sparse Attention in LLM. This combination of global self-attention with topological bias provides a balance between local structural awareness and global geometric reasoning.
>
> > W5. Lack of inverse mapping from shape to parameter space
>
> Brep2Shape focuses on the forward mapping from parameters to spatial points. The inverse direction would be essential for tasks such as shape optimization and CAE-driven design. We consider this bidirectional alignment a critically important research direction and plan to investigate it in future work.
>
> > Q1. Handling learning bias from non-uniform parametric sampling
>
> As reported in UV-Net, 89.19% of surface patches on ABC datasets exhibit chordal errors within 10⁻³ of the bounding box length, suggesting severe parametric distortion is relatively rare in typical CAD models. We further compared parametric-domain vs. spatial-domain uniform sampling: the two strategies yield nearly identical results, indicating robustness to sampling bias:
> |Sampling Strategy|Pre-training Loss|TMCAD|MFCAD++
> |-|-|-|-|
> |Spatial-domain|3.113|82.33|99.34/97.99|
> |Parametric-domain (default)|3.093|82.64|99.33/98.02|
>
> We attribute this to the pre-training objective encouraging learning of continuous geometric mappings.
>
> > Q2. Whether feature homogenization exists between face and edge streams
>
> Feature homogenization does not occur, as the two streams are supervised by different prediction targets. We further conducted a linear probing experiment: we froze the pre-trained model and attached a classification head to distinguish whether a token originates from a face or an edge. The classifier achieves **99.52% acc after one epoch**, demonstrating that the Dual Trm. maintains well-separated features.
>
> |Epoch|Acc|
> |-|-|
> |1|99.52%|
>
> > Q3. Whether higher part complexity leads to diminishing returns in sampling density
>
> We clarify that downstream performance actually improves with higher sampling density as the default because the marginal gain from m=3 to 5 is small (+0.0035 on TMCAD). Our dataset already spans 5× complexity difference (TMCAD: 75 faces vs. Fusion360Seg: 15 faces), and m=3 remains consistently effective across these datasets. For ultra-complex parts, adaptive per-primitive sampling based on local curvature is a promising extension.
>
> > Q4. Information exchange frequency and deeper cross-modal fusion strategies
>
> Face and edge streams exchange information at every layer via topology attention biases (L=6 layers), with bidirectional cues flowing through face and edge graphs. We explored two alternative deeper cross-modal fusion strategies:
>
> |Method|Pre-training Loss|TMCAD|MFCAD++|
> |-|-|-|-|
> |Q-Former|3.158|82.44|98.63/96.03|
> |Face+Edge Trm.|3.251|82.48|98.18/94.87|
> |Dual Trm.(Default)|3.093|82.64|99.34/97.99|
>
> **Face+Edge Trm.** entangles both token types and applies full self-attention, benefiting classification but degrading segmentation. **Q-Former** introduces explicit cross-attention layers between the two streams. This achieves comparable classification but underperforms on segmentation , as explicit cross-attention blurs the distinct geometric properties of faces and edges. **Topology attention** strikes the optimal balance: it preserves each stream's independence while injecting topological priors.

---

> > ### Author Rebuttal · Reviewer_Btop · 2026-04-02
> >
> > Thank you for your reply, which has solved my question. I will keep my score.

---

> > > ### Author Response · Authors · 2026-04-02
> > >
> > > We sincerely appreciate your positive feedback and are glad to hear that all your concerns have been fully resolved.
> > >
> > > Thank you for taking the time to share your thoughtful review and for your continued support.

---

### Official Review · Reviewer_tKCx · 2026-03-10

**Soundness:** 2
**Presentation:** 3
**Significance:** 3
**Originality:** 2
**Overall Recommendation:** 4
**Confidence:** 4

**Summary:**

This submission introduces Brep2Shape, which is a self-supervised learning approach for pre-training transformers on the boundary representation of CAD shapes. The key idea of the self-supervised pre-training is to learn to align the boundary representation and the point-based shape representation by predicting sampled surface points from Brep control points and topology. The authors also introduce a dual transformer backbone that processes Brep faces and edges with cross attention to topology. Experiments show the pre-training improves the performance of downstream classification and segmentation tasks.

**Compliance With Llm Reviewing Policy:**

Affirmed.

**Final Justification:**

I appreciate the authors' effort in addressing my questions. I raised the rating to weak accept.

**Key Questions For Authors:**

- Can the pre-training objective be clarified?
- Can the baseline comparisons be made clearer and fairer?
- Can the ablation study be made more complete?

**Limitations:**

Yes.

**Strengths And Weaknesses:**

Strengths:
-  The self-supervised pre-training objective, predicting surface point locations based on the Brep input is easy to implement.
- The authors performed experiments on four benchmarks, showing their pre-training learns useful representations that can benefit downstream CAD classification and segmentation tasks.
- The authors curated a Brep2Shape-250K dataset by collecting CAD models from several sources.


Weaknesses:
- The pre-training objective is straightforward and does not innovate much. It’s unclear to me whether Eq-1 is the only loss term. Fig 3 - pipeline shows there are face decoder and edge decoder. Are the Brep reconstruction losses on faces and edges not used at all? Would they improve the pre-training and downstream performance?
- It’s unclear to me why the Bezier primitives should have a fixed number of control points, considering that transformers handle sequences of varying lengths well.
- The experimental evaluation may not be complete and fair.
  - Were the baseline methods (UV-Net, AAGNet, and BRT) pre-trained on the same data (i.e., Brep2Shape-250K)? I did not find such information in the main text and the supplemental. The reported baseline performance in Tab 2 is largely borrowed from tables of BRT paper, though there are some discrepancies for UV-Net and BRT. It is important to show that the performance improvement is owing to the self-supervised pre-training scheme, rather than training on more data.
  - In the ablation study, Tab 3 only shows the training loss difference for different pre-training strategies. It’s unclear how these pre-training choices affect downstream task performance. The dual transformer backbone is claimed to be a key contribution, however, there is no ablation on the utility of incorporating topology cross attention.

---

> ### Author Rebuttal · Authors · 2026-03-30
>
> Many thanks to Reviewer tKCx for providing a detailed review and insightful questions.
> > W1 & Q1. Pre-training objective novelty & face/edge reconstruction loss usage
>
> We'd like to clarify two aspects: First, regarding novelty: we acknowledge that $\underline{Eq. 1}$ is concise, but the core novelty lies in the **task design**: predicting dense spatial points from abstract parametric Bézier control points. This forces the model to learn the nonlinear mapping from opaque polynomial coefficients to shape manifolds, involving basis function, rational normalization, rather than performing a standard reconstruction task. More importantly, our goal is not to reconstruct the original B-rep itself, but to **align abstract boundary representations with intuitive shape representations**. In this sense, reconstructing the B-rep itself may introduce undesirable side effects, while the alignment objective is the main source of novelty.
>
> Second, regarding loss: $\underline{Eq. 1}$ is the only loss used, but $\mathcal{U}$ encompasses both face and edge spatial points. As detailed in $\underline{\text{App. A.2 (Eqs. 3–5)}}$, loss is explicitly decomposed as $L_{\text{pre}}=L_{\text{face}} + L_{\text{edge}}$ where the face decoder and edge decoder predict spatial points for faces and edges.
>
> > W2. Why fixed control points per Bézier primitive
>
> The fixed-size constraint applies at the **primitive level**, not the entity level—and the two levels are handled by different components. Each B-rep entity is decomposed into a variable-length sequence of Bézier primitives, processed by the Tokenizer Transformer with a [CLS] token ($\underline{\text{Section 3.2}}$), fully leveraging the Transformer's variable-length capability.
>
> The fixed-size constraint applies within each primitive, processed by an MLP. A degree-n Bézier primitive has exactly n+1 control points; since we decompose all NURBS entities into primitives of a fixed degree ($\underline{\text{App. D}}$), this count is naturally fixed. Moreover, these control points are not a sequence, but coefficients of Bernstein basis polynomials with algebraic rather than sequential relationships, making an MLP a more suitable inductive bias than self-attention.
>
> > W3.1 & Q2 Baseline pre-training data parity & source of performance gains
>
> We would like to clarify three aspects:
>
> **(1) Baselines are not pre-trained method.** UV-Net, AAGNet, and BRT are task-specific models trained from scratch on each downstream dataset. They lack a pre-training mechanism, making it infeasible to pre-train them on Brep2Shape-250K. In contrast, Brep2Shape is pre-trained for 100 epochs and then fine-tuned for 100 epochs, while baselines follow their original setting of 350 training epochs.
>
> **(2) Reproduction details.** We reproduced UV-Net and BRT on all benchmarks. For AAGNet, due to compatibility issues, we used the originally reported numbers where our reproduction was not feasible.
>
> **(3) Gains come from pre-training, not data volume.** In the data scaling experiment ($\underline{\text{Fig. 4}}$) , Brep2Shape with only 25K pre-training samples already surpasses BRT on most tasks. In the limited data study ($\underline{\text{Fig. 5}}$), all methods use identical labeled data, and Brep2Shape consistently outperforms all baselines across all labeling regimes. These confirm that the improvement stems from the pre-training scheme, not data volume.
>
> > W3.2 & Q3 Missing downstream ablations for pre-training strategies & topology attention utility
>
> We have conducted additional downstream evaluation for the pre-training strategy ablations:
>
> |Method|Face Loss|Edge Loss|TMCAD|MFCAD++|
> |-|-|-|-|-|
> |Pre-train Tokenizers Only|2.142|1.287|80.40|99.16/97.56|
> |Dual Trm. w/o Edge-Level Supervision|1.872|–|81.43|99.18/97.56|
> |Brep2Shape (Default)|**1.855**|**1.238**|**82.64**|**99.33/98.02**|
>
> Both components contribute meaningfully to downstream performance: removing the Dual Trm causes a 2.24% drop in TMCAD; removing edge-level supervision leads to a 1.21% drop.
>
> **Regarding topology attention:** This ablation is already presented in $\underline{\text{Tab. 4}}$, where Brep2Shape consistently outperforms all variants on both TMCAD and MFCAD++. We would like to clarify that our topology attention is not cross-attention, but self-attention augmented with attention biases derived from the topological graph through which cross-stream information flows. To further validate, we experimented with Q-Former-style explicit cross-attention:
>
> |Backbone|Pre-training Loss|TMCAD|MFCAD++|
> |-|-|-|-|
> |Q-Former|3.158|82.44|98.63/96.03|
> |Dual Trm. w/ Std. Attn.|3.203|81.77|98.53/95.82|
> |Dual Trm. (Default)|**3.093**|**82.64**|**99.34/97.99**|
>
> Q-Former achieves comparable classification but underperforms on segmentation. We hypothesize that explicit cross-attention disrupts entity-specific representations critical for face-level prediction, while our topology attention preserves stream independence and achieves the best overall balance.

---

> > ### Author Rebuttal · Reviewer_tKCx · 2026-04-02
> >
> > Thank you for the rebuttal, which addressed my questions.

---

> > > ### Author Response · Authors · 2026-04-02
> > >
> > > Thank you very much for your positive feedback and for marking our concerns as fully resolved. We truly appreciate your time and thoughtful evaluation.
> > >
> > > If there are any further questions or aspects that you would like us to clarify, we would be more than happy to provide additional details.
> > >
> > > If you feel that our responses have sufficiently addressed the concerns, we would sincerely appreciate it if you could kindly reconsider your evaluation.
> > >
> > > Thank you again for your support.

---

### Official Review · Reviewer_YzsX · 2026-03-14

**Soundness:** 3
**Presentation:** 3
**Significance:** 4
**Originality:** 3
**Overall Recommendation:** 5
**Confidence:** 4

**Summary:**

This paper presents Brep2Shape, a self-supervised pre-training method for aligning abstract boundary representations with explicit shape representations in CAD models. This alignment task is intended to bridge the gap between mathematically precise boundary descriptions and intuitive geometric perception. To enhance this alignment and maintain topological consistency, a Dual Transformer backbone with topology-aware attention is proposed.

To train the Brep2Shape model, this work also curates Brep2Shape-250k, consisting of 250,000 B-rep models from multiple
public and industrial sources.
Experiments across several large-scale benchmarks demonstrate the scalability, accuracy, and data efficiency of Brep2Shape, outperforming state-of-the-art baselines on downstream tasks.

**Compliance With Llm Reviewing Policy:**

Affirmed.

**Key Questions For Authors:**

1. The computational cost for pre-training is explicitly detailed as 38 GPU-hours on an A100. Could the authors provide a comparison of inference speeds or parameter counts against primary baselines like BRT or UV-Net to contextualize the method's deployment efficiency?
2. How does the framework handle edge cases where Bézier control points become degenerate or highly overlapping, which frequently occurs in poorly modeled or imported CAD geometry?
3. The influence of topology attention is showed in ablation study. Could the authors provide the intermediate results of the attention map making its significance more explicit?
4. The tSNE visualization of faces in Fig 6 (b) is quite abstract. Could the authors visualize the face examples that two nearby points and distant points in the feature space?

**Limitations:**

Yes. The authors discussed the limitations such as that the fixed-order decomposition can introduce approximation biases unsuitable for ultra-precision applications like aerospace manufacturing, the lack of established edge-level tasks for evaluation and the current model's limitation in handling multi-body assemblies and joint constraints.

**Strengths And Weaknesses:**

**Strengths:**
1. **Originality:** Bridging the analytical exactness of continuous models with the spatial intuition of discrete models via self-supervised alignment is a highly original and effective approach. The design of the Dual Transformer, specifically its use of an edge graph acting as the dual of the face graph to inject bidirectional topological priors as attention biases, is a novel structural adaptation for B-reps.

2. **Significance:** As B-rep is the de facto industry standard for 3D modeling in engineering, overcoming its inherent deep learning challenges carries high practical relevance. The curation of Brep2Shape-250k from multiple data sources, utilizing only raw B-rep geometry independent of downstream labels, provides a highly scalable foundation for future CAD research.

3. The paper is **technically robust**, featuring extensive experimental validation across four diverse datasets. The authors provide a rigorous analysis of both data scaling (showing performance gains up to 250k samples) and model scaling (increasing layers from 2 to 12), proving the architecture's capacity to handle complex geometric information. Ablation studies, scaling experiments, detailed backbone analysis, and low-data regime studies add empirical depth and credibility.

4. **Presentation:** The paper is well-structured and uses clear visualizations to illustrate the representation gap and the model's predictive accuracy.

**Weakness**

1. The evaluation is focused on face-level tasks (classification, segmentation). Edge-level or assembly-level prediction is not addressed or evaluated, despite design claims of topological alignment (Appendix F). This restricts validation of generality.
2. No evaluation on the reconstruction and generation tasks.

---

> ### Author Rebuttal · Authors · 2026-03-30
>
> We would like to sincerely thank Reviewer YzsX for providing a detailed review and insightful questions.
>
> > W1 & W2 Edge-level & assembly-level prediction & reconstruction & generation tasks
>
> We address the three aspects raised. **(i) Edge-level evaluation.** Edge-level supervision produces meaningful representations. Explicit edge-level benchmarks are currently absent in the community, which we hope our work can help motivate. **(ii) Assembly-level evaluation.** We acknowledge this limitation (noted in $\underline{\text{App. F (iii)}}$). Assembly-level reasoning could be approached by a hierarchical graph with cross-body attention on top of the Dual Transformer. **(iii) Generative tasks.** We clarify that our claim is generalizable, not general representations. Just as MAE excels at recognition but requires adaptation for generation, our reconstruction-based pre-training naturally aligns with understanding tasks. Extending to generative tasks would require complementary objectives, an exciting future direction.
>
> > Q1. Inference efficiency comparison with baselines
>
> We provide the comparison below (all measured on the same A100 GPU):
>
> |Method|Params(M)|Latency(ms/sample)|
> |-|-|-|
> |UV-Net|1.3|6.73|
> |BRT|2.1|8.43|
> |Brep2Shape|2.1|8.62|
>
> **Deployment efficiency**: Brep2Shape and BRT share nearly identical parameter counts and inference latency (only 2.3% slower). This negligible overhead is well justified by consistently superior accuracy. Compared to UV-Net, the modest increase in parameters and latency is offset by substantial gains ($\underline{\text{eg. 83.77\\% vs 66.47\\% IoU on Fusion360Seg}}$).
>
> **Training efficiency**: Brep2Shape requires only **100 fine-tuning epochs** to outperform baselines trained from scratch for **350 epochs**. Once the one-time pre-training cost (38 A100-hours) is amortized across downstream tasks, per-task training cost is significantly reduced. Overall, Brep2Shape introduces virtually no deployment overhead while offering clear advantages in both accuracy and training cost.
>
> > Q2. Handling degenerate Bézier control points
>
> Our framework handles this through multiple layers of defense: **(1) Data preprocessing.** We filter out severely degenerate entities (e.g., zero-area faces or fully collapsed control point nets). For near-overlapping control points within floating-point precision, the pipeline remains numerically stable. **(2) Architectural robustness.** The topology attention enables information flow from neighboring faces and edges, providing robustness to local degeneracies — a partially degenerate entity can leverage geometric semantics from its topological neighbors, mitigating the impact of poor local geometry on the overall representation quality. **(3) Empirical evidence.** TMCAD dataset comprises real-world models from the Internet, with far less controlled geometric quality than datasets like MFCAD++. Brep2Shape achieves SOTA on TMCAD, demonstrating robustness to imperfect real-world geometries.
>
> > Q3. Attention map visualization for topology attention
>
> We will include attention map visualizations in the revised version. We provide a qualitative analysis of the observed patterns: on some heads, both mechanisms produce similar content-driven interactions. However, a key distinction emerges on other heads: topology attention incorporates topological priors via cross-stream attention biases, producing structured patterns closely resembling the B-rep adjacency matrix and concentrating on topologically neighboring entities. In contrast, standard attention heads produce more diffuse distributions, as they must discover structural relationships purely from data. This aligns with the performance gap in $\underline{\text{Tab. 4}}$, where topology attention yields notably better segmentation — a task critically depending on local adjacency and boundary consistency.
>
> > Q4. Concrete face examples for t-SNE visualization
>
> We appreciate this suggestion and provide two quantitative analyses to complement the t-SNE visualization in $\underline{\text{Fig. 6(b)}}$. **(1) Cosine similarity analysis.**  We randomly selected 20 faces and computed the cosine similarity of their embeddings against all other faces. For each selected face, we examined the top-10% most similar and bottom-10% least similar neighbors and measured the label agreement rate. **(2) Linear probing.** We froze each representation and trained a single-layer MLP classifier. Results are summarized below:
>
> |Repr.|Discrim. Score|Linear Probe Acc.|
> |-|-|-|
> |Boundary Repr.|14.86%|70.35%|
> |Shape Repr.|20.10%|73.53%|
> |Brep2Shape Repr.|**23.33%**|**77.33%**|
>
> Both analyses confirm that Brep2Shape representations exhibit superior discriminability compared to either boundary or shape representations alone, validating that the alignment objective produces more semantically meaningful feature spaces. We will include face-level nearest/farthest neighbor visualizations in the revised version.

---

> > ### Author Rebuttal · Reviewer_YzsX · 2026-04-06
> >
> > As the authors full resolved my concerns about the inference efficiency and minor issues of the visualization, I keep my score as "accept".

---

> > > ### Author Response · Authors · 2026-04-06
> > >
> > > We sincerely appreciate your generous feedback. It is most gratifying to learn that all matters of concern have been resolved to your complete satisfaction. We are grateful for the time you have taken to share your valuable insights, and we remain deeply appreciative of your continued trust and patronage.

---

### Decision · Program_Chairs · 2026-04-30

**Decision:**

Accept (regular)

**Comment:**

This paper aims to reduce the gap between continuous and discrete methods to represent B-rep CAD models for deep learning. Although reviewers raised concerns about limited evaluation and potential unfair comparison with a stronger pre-training scheme, the authors provided an extensive rebuttal that addressed these concerns. AC notes that the new CAD dataset would be valuable to the community. Based on the positive reviews, AC recommends acceptance of the paper. AC requests that authors polish the paper based on the additional results and discussions during the rebuttal phase.